# Design principles for inflammasome inhibition by pyrin-only-proteins

Shuai Wu[1], Archit Garg[1†], Zachary Mazanek[1†‡], Gretchen Belotte[1], Jeffery J Zhou[1], Christina M Stallings[1], Jacob Lueck[1], Aubrey Roland[2], Michael A Chattergoon[2§], Jungsan Sohn[1,3,4]*

[1]Department of Biophysics and Biophysical Chemistry, Johns Hopkins University School of Medicine, Baltimore, United States; [2]Division of Infectious Diseases, Johns Hopkins University School of Medicine, Baltimore, United States; [3]Division of Rheumatology, Johns Hopkins University School of Medicine, Baltimore, United States; [4]Department of Oncology, Johns Hopkins University School of Medicine, Baltimore, United States

*For correspondence:
jsohn@jhmi.edu

[†]These authors contributed equally to this work

Present address: [‡]Georgiamune, Gaithersburg, United States; [§]Vir Biotechnology, San Francisco, United States

**Abstract** Inflammasomes are filamentous signaling platforms essential for host defense against various intracellular calamities such as pathogen invasion and genotoxic stresses. However, dysregulated inflammasomes cause an array of human diseases including autoinflammatory disorders and cancer. It was recently identified that endogenous pyrin-only-proteins (POPs) regulate inflammasomes by directly inhibiting their filament assembly. Here, by combining Rosetta in silico, in vitro, and in cellulo methods, we investigate the target specificity and inhibition mechanisms of POPs. We find here that POP1 is ineffective in directly inhibiting the central inflammasome adaptor ASC. Instead, POP1 acts as a decoy and targets the assembly of upstream receptor pyrin-domain (PYD) filaments such as those of AIM2, IFI16, NLRP3, and NLRP6. Moreover, not only does POP2 directly suppress the nucleation of ASC, but it can also inhibit the elongation of receptor filaments. In addition to inhibiting the elongation of AIM2 and NLRP6 filaments, POP3 potently suppresses the nucleation of ASC. Our Rosetta analyses and biochemical experiments consistently suggest that a combination of favorable and unfavorable interactions between POPs and PYDs is necessary for effective recognition and inhibition. Together, we reveal the intrinsic target redundancy of POPs and their inhibitory mechanisms.

## Editor's evaluation

This important work proposes general principles by which a central class of inflammatory mediators called inflammasomes are regulated by "pyrin only" proteins. The authors employ a convincing combination of in silico and biochemical measurements in support of their model. The work is of broad interest to those interested in the biochemical regulation of inflammatory signaling pathways.

## Introduction

Inflammasomes are filamentous signaling platforms integral to host innate defense against a wide range of intracellular catastrophes, which include ionizing irradiation, genotoxic chemicals, and pathogen invasion (*Broz and Dixit, 2016*; *Zheng et al., 2020*). However, persisting inflammasome activities lead to several human maladies including numerous autoinflammatory disorders, cancer, and even severe COVID-19 (*Karki et al., 2017*; *Tartey and Kanneganti, 2020*; *Vora et al., 2021*). Thus, understanding how inflammasome assemblies are regulated at the molecular level can provide key

insights into developing strategies for preventing and treating various diseases (*Broz and Dixit, 2016*; *Karki et al., 2017*; *Tartey and Kanneganti, 2020*; *Vora et al., 2021*; *Zheng et al., 2020*).

Inflammasomes transduce signals by sequentially assembling filamentous oligomers, with multiple initial pathways progressively converging at downstream assemblies (*Broz and Dixit, 2016*; *Kagan et al., 2014*; *Lu and Wu, 2015*; *Zheng et al., 2020*). For instance, an array of molecular signatures arising from various pathogenic conditions induces the oligomerization of inflammasome receptors, resulting in filament assembly by their pyrin-domains (PYDs; e.g. viral nucleic acids, reactive oxygen species, specific lipids from damaged mitochondria, and disruption of the trans-Golgi network) (*Andreeva et al., 2021*; *Fernandes-Alnemri et al., 2009*; *Hornung et al., 2009*; *Iyer et al., 2013*; *Roberts et al., 2009*; *Zhong et al., 2013*); PYDs are ~100 amino acid (a.a.) 6-helix bundle proteins that belong to the death-domain (DD) family often found in apoptotic and inflammatory signaling pathways (*Park et al., 2007*). The upstream PYD oligomers then induce the filament assembly by the PYD of the central adaptor ASC (ASC$^{PYD}$), resulting in oligomerization/filamentation of its CARD (ASC: apoptosis-associated speck-forming protein-containing caspase-recruiting domain [CARD]) (*Broz and Dixit, 2016*; *Kagan et al., 2014*; *Lu et al., 2014*; *Lu and Wu, 2015*). Finally, ASC$^{CARD}$ oligomers recruit pro-caspase-1 and induce its filament assembly, activating the enzyme by proximity-induced auto-proteolysis (*Broz and Dixit, 2016*; *Kagan et al., 2014*; *Lu et al., 2014*; *Lu and Wu, 2015*). Caspase-1 executes two key innate immune responses, namely the cleavage/maturation of pro-inflammatory cytokines such as interleukin-1β (IL-1β) and IL-18, and the initiation of pyroptosis (*Broz and Dixit, 2016*; *Kagan et al., 2014*; *Lu et al., 2014*; *Lu and Wu, 2015*).

A hallmark of inflammasome assembly is its binary (on-or-off) nature (*Cai et al., 2014*; *Franklin et al., 2014*; *Matyszewski et al., 2018*; *Shen et al., 2021*). That is, once assembled, inflammasomes do not dissociate (*Franklin et al., 2014*; *Matyszewski et al., 2018*). Moreover, multiple positive feedback loops between upstream receptors and ASC not only bolster the assembly, but also result in prion-like self-perpetuation (*Cai et al., 2014*; *Matyszewski et al., 2018*). Such an inherently irreversible assembly mechanism in turn would necessitate extrinsic factors to prevent persistent activities. Indeed, mammalian pyrin-only-proteins (POPs) have emerged as major inhibitors of inflammasomes (*de Almeida et al., 2015*; *Khare et al., 2014*; *Periasamy et al., 2017*; *Ratsimandresy et al., 2017*), functioning analogous to CARD-only proteins (COPs) that interfere with the oligomerization/activation of pro-caspases (*Devi et al., 2020*; *Indramohan et al., 2018*; *Lu et al., 2016*). It has been proposed that the target specificities of POPs are dictated by their a.a. sequence homologies to inflammasome PYDs (*Devi et al., 2020*; *Indramohan et al., 2018*). For example, POP1 is most homologous to ASC$^{PYD}$ (65% sequence identity; *Figure 1—figure supplement 1A*) and thought to directly inhibit the nucleation of the ASC$^{PYD}$ filament (*de Almeida et al., 2015*). POP2 shares 68% sequence identity to the PYD of Nod-like receptor containing a PYD-2 (NLRP2$^{PYD}$; *Figure 1—figure supplement 1B*); the primary target of POP2 is thought to be ASC, but it is also implicated in inhibiting the absent-in-melanoma-2 (AIM2) receptor (*Periasamy et al., 2017*; *Ratsimandresy et al., 2017*). Finally, POP3 is most similar to AIM2$^{PYD}$ (67% sequence identity, *Figure 1—figure supplement 1C*) and targets AIM2-like receptors (ALRs, e.g. AIM2 and interferon inducible protein 16 [IFI16]) (*Khare et al., 2014*).

Inflammasome filaments are highly ordered supra-structures that entail at least two distinct steps for assembly: rate-limiting nucleation followed by elongation (*Kagan et al., 2014*; *Lu et al., 2014*; *Lu and Wu, 2015*; *Matyszewski et al., 2018*). Moreover, although the PYDs of upstream receptors do not display significant a.a. sequence homologies (*Kagan et al., 2014*; *Lu et al., 2014*; *Lu and Wu, 2015*), they all assemble into structurally congruent helical filaments and signal through the common ASC adaptor, suggesting a degenerate-code-like recognition mechanism (*Hochheiser et al., 2022a*; *Kagan et al., 2014*; *Lu et al., 2014*; *Lu and Wu, 2015*; *Matyszewski et al., 2021*; *Shen et al., 2019*). At present, little is known about how POPs selectively target and regulate the assembly of such diverse yet homologous supramolecular structures. This is because the current understanding on the mechanism of inhibition by POPs remains entirely inferred from indirect measurements and phenotypic outcomes (*de Almeida et al., 2015*; *Khare et al., 2014*; *Periasamy et al., 2017*; *Ratsimandresy et al., 2017*).

Here, by combining in silico, in cellulo, and in vitro methods, we delineate the target specificity and inhibition mechanisms of human POPs. We find that POP1 is a poor inhibitor of ASC and impedes the assembly of upstream receptor filaments instead (e.g. AIM2, IFI16, NLRP3, NLRP6). POP2 not only suppresses the nucleation of ASC, but also interferes with the assembly of multiple upstream

receptors. Finally, in addition to potently suppressing the assembly of ALR and NLRP filaments (e.g. elongation of AIM2$^{PYD}$), POP3 suppresses the nucleation of ASC. Our results indicate that a combination of favorable and strongly unfavorable interactions is necessary for POPs to inhibit the assembly of PYD filaments. Together, we propose that, instead of being dictated by a.a. sequence homology, degenerate-code-like target selection and inhibition mechanisms underpin the regulation of inflammasome assembly by POPs.

## Results

### Rosetta interface analyses suggest broad target specificities of POPs

Several inflammasome receptor PYDs signal through ASC$^{PYD}$ although their primary a.a. sequences vastly differ (*Broz and Dixit, 2016*; *Kagan et al., 2014*; *Lu and Wu, 2015*). Such a functional redundancy among different PYDs in turn suggests that sequence homology may not dictate the target specificity of POPs. To elucidate how POPs recognize and regulate the assembly of different PYD filaments, we first implemented Rosetta-based in silico approach that we had recently developed to define the directionality of the AIM2-ASC inflammasome (*Matyszewski et al., 2021*). Briefly, PYDs assemble into helical filaments in which each protomer provides six unique protein-protein interaction surfaces (e.g. *Figure 1A*, 'Type' 1a/b, 2a/b, and 3a/b) (*Lu et al., 2014*; *Lu and Wu, 2015*). As we had done before (*Matyszewski et al., 2021*), we created a honeycomb-like side view of PYD filaments in which the middle protomer makes all six required contacts for filament assembly (*Figure 1A*). We then calculated Rosetta interface energies (ΔGs) for each PYD filament (e.g. *Figure 1B*, left), and also determined the ΔGs for POP•PYD interactions by replacing the center protomer with each POP (e.g. *Figure 1B*, three honeycombs on the right). Of note, we decided to conduct our in silico and subsequent biochemical experiments on tractable inflammasome components with well-known biological significances such as ASC, AIM2, IFI16, NLRP3, and NLRP6 (*Broz and Dixit, 2016*; *Hochheiser et al., 2022a*; *Kerur et al., 2011*; *Lu et al., 2014*; *Lu and Wu, 2015*; *Matyszewski et al., 2018*; *Matyszewski et al., 2021*; *Morrone et al., 2015*; *Morrone et al., 2014*; *Shen et al., 2019*).

We found previously that individual AIM2$^{PYD}$ and ASC$^{PYD}$ filaments assemble bidirectionally (i.e. extending from both top and bottom surfaces), with each pair of filament interface types contributing symmetric ΔGs (e.g. both Type 1a and Type 1b surface show ΔG=−22.1 for ASC$^{PYD}$•ASC$^{PYD}$ in *Figure 1B*; *Matyszewski et al., 2021*). The interface analysis results from other inflammasome PYDs also showed similar symmetric energy landscapes (*Figure 1B–F*, left), suggesting that bidirectional assembly is universal to all PYD filaments. We next noted that the overall ΔGs for individual PYD filaments were more favorable than those from any putative POP•PYD interactions, which in turn suggested that excess POPs might be necessary to inhibit the assembly of inflammasome PYDs (e.g. the sum of ASC$^{PYD}$•ASC$^{PYD}$ interactions on the top half yields ΔG=−35.9, while that of POP1•ASC$^{PYD}$ is −24.7; *Figure 1B–F*). POP1 showed more favorable overall ΔGs for ASC$^{PYD}$ compared to POP2 or POP3 (*Figure 1B*), seemingly supporting the previous report (and sequence homology) that POP1 likely binds ASC (*de Almeida et al., 2015*). However, although it was reported that POP2 can inhibit ASC$^{PYD}$ more potently than POP1 in vivo (*Ratsimandresy et al., 2017*), its ΔGs were less favorable (i.e. *Figure 1B*: POP1•ASCP$^{YD}$ = −24.7 vs. POP2•ASC = −0.8 on the top half). POP3, on the other hand, appeared to interact with ASC as favorably as POP1 at the bottom interfaces (*Figure 1B*; ΔG=−34.3 for POP1•ASC$^{PYD}$ vs. ΔG=−34.6 for POP3•ASC$^{PYD}$), suggesting that it could also inhibit the central adaptor.

For AIM2$^{PYD}$, all three POPs showed comparable overall ΔGs on the bottom half (*Figure 1C*), which suggested that each of them could target AIM2. IFI16 favored POP3 the most (*Figure 1D*; e.g. ΔG=−9.6 for POP3•IFI16$^{PYD}$ on the top half); however, the other two POPs still showed more favorable ΔGs than homotypic IFI16$^{PYD}$•IFI16$^{PYD}$ interactions on at least one individual interface (*Figure 1D*; e.g. Type 2a for POP1, Type 2b for POP2, and Type 1b for POP3). These results in turn suggested that not only POP3, but POP1 and POP2 might also recognize IFI16. It has been speculated that POPs interfere with the recruitment of ASC by the upstream receptors (*de Almeida et al., 2015*; *Devi et al., 2020*; *Indramohan et al., 2018*; *Periasamy et al., 2017*; *Ratsimandresy et al., 2017*); however, it remains unknown whether they do so by directly inhibiting NLRPs, which are the major class of inflammasome receptors. Our Rosetta analyses here suggest that POP1 could interact with NLRP3$^{PYD}$ on the top half (*Figure 1E*; ΔG=−31.4 for NLRP3$^{PYD}$•NLRP3$^{PYD}$ vs. ΔG=−30.5 for POP1•NLRP3$^{PYD}$).

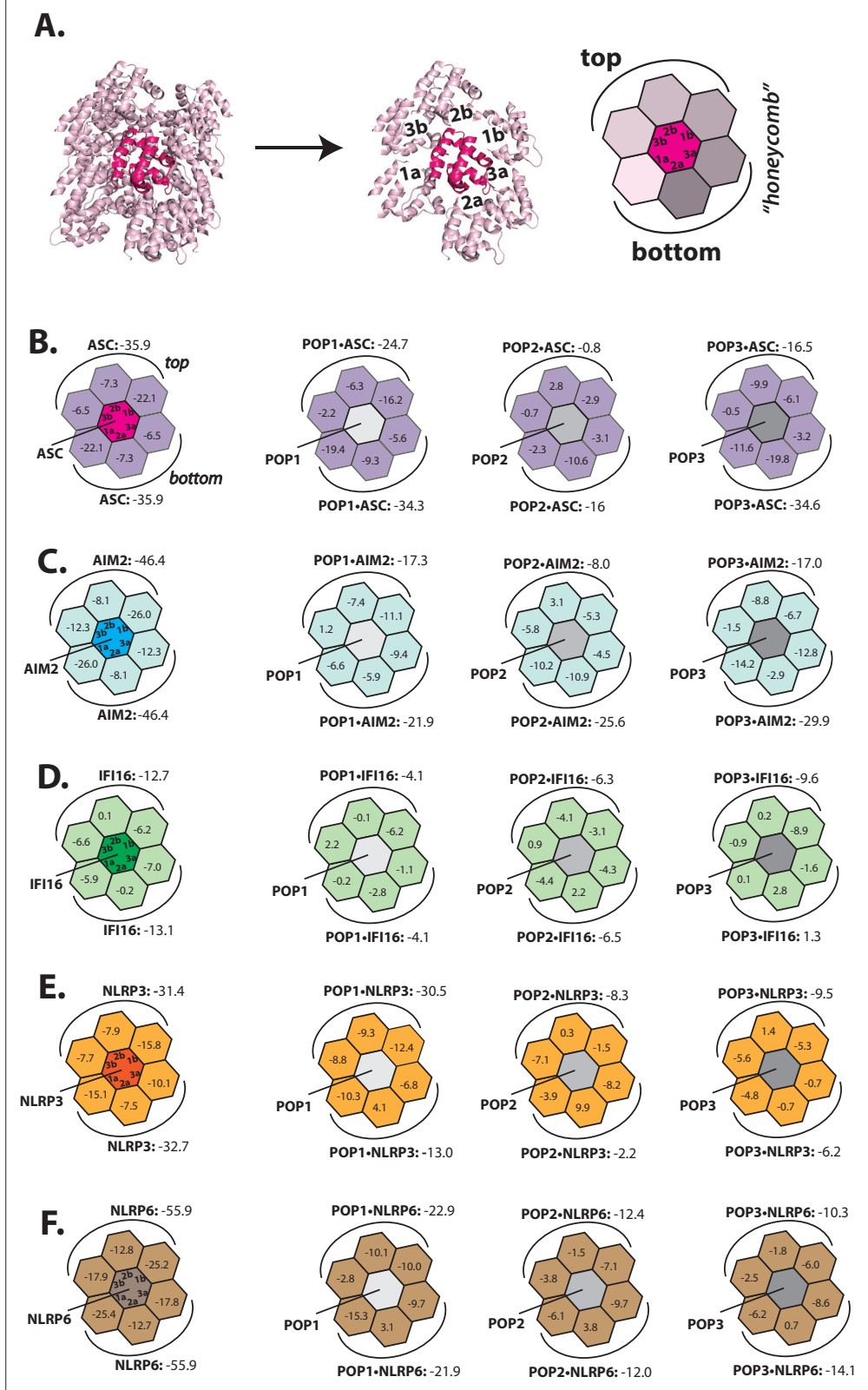

**Figure 1.** Rosetta in silico analyses of putative pyrin-only-protein (POP)•pyrin-domain (PYD) interactions. (**A**) The 'side view' of the ASC[PYD] filament (PDB: 3j63). The center magenta protomer makes all six unique contacts with surrounding pink protomers for assembly. Each surrounding protomer is colored in a different shade of pink in the 'honeycomb' diagram. (**B–F**) Rosetta interface energy scores (ΔGs, Rosetta energy score, reu) at individual filament

*Figure 1 continued on next page*

*Figure 1 continued*

interfaces for homotypic assemblies (left) and putative interactions with POPs (right). Each hexagon represents a PYD or POP monomer. The sum of ΔGs at the top and bottom half is also listed. The honeycombs were generated based on their respective cryo-EM structures except for IFI16$^{PYD}$ whose filament structure is unknown. We generated a homology model of IFI16$^{PYD}$ filament based on the GFP-tagged AIM2 filament (PDB: 6mb2), which produced more favorable ΔGs than the one generated from the tagless-AIM2$^{PYD}$ filament (PDB: 7k3r; *Figure 1—figure supplement 2*).

The online version of this article includes the following figure supplement(s) for figure 1:

**Figure supplement 1.** Amino acid sequence alignments of POPs.

**Figure supplement 2.** FI16$^{PYD}$ honeycombs based on two different homology models.

However, the interactions between POP2/3 and NLRP3$^{PYD}$ appeared much less favorable (*Figure 1E*). Similar to NLRP3$^{PYD}$, POP1 again showed more favorable energy scores toward NLRP6$^{PYD}$ than POP2/3 (*Figure 1F*). Overall, although these results appear to rationalize some of the proposed target specificities of POPs (*Devi et al., 2020*; *Indramohan et al., 2018*), they also suggest confounding interaction and recognition mechanisms, thus warranting further investigations via biochemical approaches.

## Mechanisms of ASC inhibition by POPs

Previous investigations on POP•PYD interactions have predominantly relied on in cellulo downstream signaling activities and in vivo phenotypes (*de Almeida et al., 2015*; *Khare et al., 2014*; *Periasamy et al., 2017*; *Ratsimandresy et al., 2017*). Although establishing the physiological relevance of POPs, these studies have left large voids in understanding their target selection and inhibition mechanisms. Because our in silico analyses did not immediately yield clear explanations for such questions, we set out to test POP•PYD interactions using more direct in cellulo and in vitro methods, focusing on ASC first. When ectopically expressed in HEK293T cells, C-terminally mCherry-tagged ASC$^{PYD}$ forms filaments and full-length ASC (ASC$^{FL}$) forms puncta (*Matyszewski et al., 2021*). Importantly, HEK293T cells do not contain any endogenous inflammasome components or POPs, providing an ideal cellular system for directly tracking their interactions (*de Almeida et al., 2015*; *Matyszewski et al., 2021*; *Shi et al., 2016*). Here, we first tested whether co-transfecting C-terminally eGFP-tagged POPs hinders the oligomerization of ASC. Compared to co-transfecting eGFP alone, POP1-eGFP only marginally inhibited the filament assembly of ASC$^{PYD}$-mCherry (≤20% suppression vs. eGFP control; *Figure 2A–B*). By contrast, co-transfecting POP2-eGFP or POP3-eGFP significantly reduced the amount of ASC$^{PYD}$-mCherry filaments, with POP2 being more effective (*Figure 2A–B*; note more diffused mCherry signals and reduction in linear filaments in the presence of POP2 and POP3 in *Figure 2A*). Furthermore, POP1 reduced the number of ASC$^{FL}$ puncta by ~30%, yet POP2 and POP3 were again more effective in preventing punctum formation (up to ~80% reduction, *Figure 2C–D*). Monitoring oligomerization of mCherry-labeled ASC in the presence of untagged POPs also corroborated our observations with eGFP-tagged POPs (*Figure 2—figure supplement 1*). These results suggest that POP1 may not directly target ASC$^{PYD}$. Moreover, unlike COPs that co-assemble with CARDs into filaments (*Lu et al., 2016*), our observations indicate that POPs suppress filament assembly altogether.

Our observations strongly indicate that POP1 does not directly target ASC assembly. To further test our in cellulo results, we then generated recombinant POPs to investigate their inhibitory mechanisms. POP1 behaved as a monomer without forming filaments or higher-order species in our hands (*Figure 2—figure supplement 2A–B*). On the other hand, recombinant POP2 and POP3 were prone to aggregation/degradation during purification and required an N-terminal maltose-binding protein (MBP) tag to obtain intact proteins (*Figure 2—figure supplement 2A*). Cleaving MBP via tobacco etch virus protease (TEVp) indicated that POP2 and POP3 form elongated oligomers with undefined structures (*Figure 2—figure supplement 2B*).

We incorporated recombinant POPs into our well-established polymerization assay in which we track the Förster resonance energy transfer (FRET) ratio between a 1:1 mixture of donor- and acceptor-labeled PYDs (*Matyszewski et al., 2018*; *Matyszewski et al., 2021*; *Mazanek and Sohn, 2019*); the auto-assembly of individual PYDs is suppressed by an N-terminal MBP tag and polymerization is triggered by cleaving MBP via TEVp. Of note, our assay displays two distinct phases of filament assembly, namely the rate-limiting nucleation (initial lag; double/triple-headed arrows in *Figure 2E*) followed by

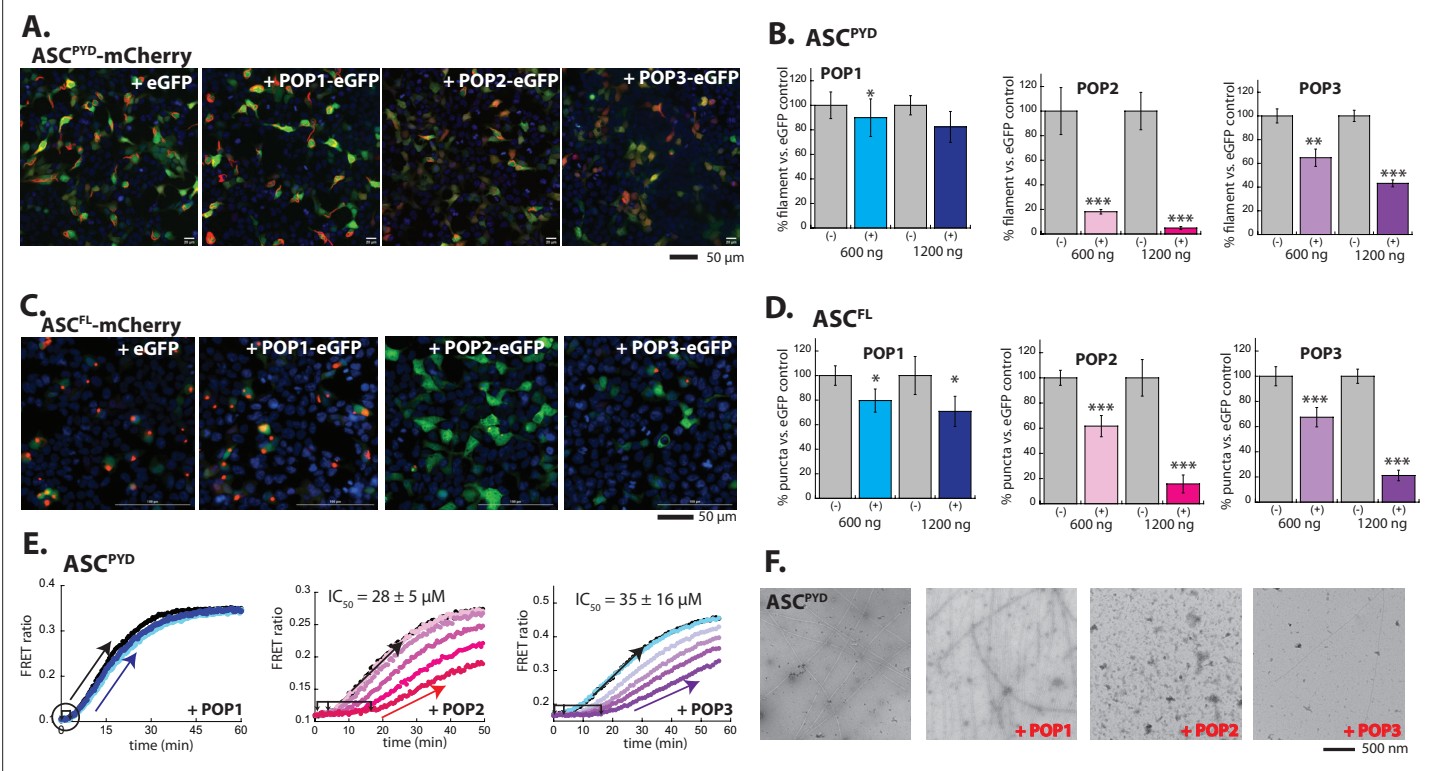

**Figure 2.** POP1 does not directly inhibit apoptosis-associated speck-forming protein-containing caspase-recruiting domain (ASC). (**A**) Sample fluorescent microscope images of HEK293T cells co-transfected with mCherry-tagged ASC$^{PYD}$ (300 ng; crimson) plus eGFP alone or eGFP-tagged pyrin-only-proteins (POPs) (1200 ng; green). Blue: DAPI. (**B**) The relative amounts of ASC$^{PYD}$-mCherry filaments (300 ng plasmid) in HEK293T cells when co-transfected with POP-eGFP (+) or eGFP alone (-) (600 and 1200 ng plasmids). n≥4. *: p≤0.05, **: p≤0.01; ***: p≤0.001, hereafter. (**C**) Sample fluorescent microscope images of HEK293T cells co-transfected with mCherry-tagged ASC$^{FL}$ (300 ng; crimson) plus eGFP alone or eGFP-tagged POPs (1200 ng; green). Blue: DAPI. (**D**) The relative amounts of ASC$^{FL}$-mCherry puncta (300 ng plasmid) in HEK293T cells when co-transfected with POP-eGFP (+) or eGFP (-) (600 and 120 ng plasmids). N≥4. (**E**) Time-dependent increase in Förster resonance energy transfer (FRET) signals of a donor- and acceptor-labeled ASC$^{PYD}$ (2.5 μM total, black circle) was monitored with increasing concentrations POP1 (50 and 150 μM), POP2 (3.3, 6.7, 13.3, 26.7, and 40 μM), or POP3 (3.25, 7.5, 15, 20, and 30 μM); darker shades correspond to increasing POP concentrations. Two- and three-headed arrows indicate the increase in apparent nucleation time (or lack thereof). Arrows pointing upper right directions indicate the change (or lack thereof) in the elongation phase in the presence of the highest POP concentrations used. Data shown are representatives of at least three independent measurements (IC$_{50}$s are average values of these experiments. N=3). (**F**) Negative-stain electron microscopy (nsEM) images of ASC$^{PYD}$ filaments (2.5 μM) in the presence and absence of POP1 (150 μM), POP2 (40 μM), or POP3 (30 μM).

The online version of this article includes the following source data and figure supplement(s) for figure 2:

**Source data 1.** Data values used in plots from *Figure 2* and *Figure 2—figure supplements 1–2* and a folder containing uncropped gel images used in *Figure 2—figure supplement 2*.

**Figure supplement 1.** Inhibition of ASC assembly by pyrin-only-proteins (POPs).

**Figure supplement 2.** Preparation and characterization of recombinant POPs.

**Figure supplement 3.** Rosetta interface analysis results showing favorable (ΔΔG≤3.5 reu, blue dots) and unfavorable (ΔΔG≥10.0 reu, red dots) interactions between ASC$^{PYD}$ and pyrin-only-proteins (POPs).

elongation (exponential/linear phase; single-headed arrows pointing to the upper right-hand corner in *Figure 2E*; *Matyszewski et al., 2018*; *Matyszewski et al., 2021*; *Mazanek and Sohn, 2019*).

Consistent with our cellular imaging assays (*Figure 2A–B*), POP1 did not affect the oligomerization kinetics or maximum FRET efficiency (amplitude) of ASC$^{PYD}$ up to the highest concentration we could achieve (*Figure 2E*; 150 μM POP1 vs. 2.5 μM ASC$^{PYD}$). Moreover, no FRET signals were detected between donor-labeled POP1 and acceptor-labeled ASC$^{PYD}$ (*Figure 2—figure supplement 2C*). Additionally, the presence of POP1 did not affect the formation of ASC$^{PYD}$ filaments when visualized with negative-stain electron microscopy (nsEM) (*Figure 2F*). By contrast, both POP2 and POP3 prolonged the initial lag phase of ASC$^{PYD}$ polymerization in a dose-dependent manner (up to ~15 min delay in

nucleation; *Figure 2E*), while also moderately affecting the elongation phase (≤20% reduction in the slope; *Figure 2E*). Estimating the polymerization half-times at different POP concentrations indicated that POP2 and POP3 are similarly effective in suppressing the oligomerization of ASC$^{PYD}$ (*Figure 2E*, IC$_{50}$s). When visualized via nsEM, no ASC$^{PYD}$ filaments were detected in the presence of POP2, and only a few filaments were seen with POP3 (*Figure 2F*). These results suggest that even if ASC$^{PYD}$s form oligomers (rise in FRET signals in *Figure 2E*), most of them fail to assemble into intact filaments in the presence of POP2/3. Moreover, the complete absence of any ASC$^{PYD}$ filaments in the presence of POP2 is consistent with our cellular experiments in which POP2 was most potent in inhibiting the central adaptor (*Figure 2B and F*). Together, our in cellulo and in vitro experiments consistently indicate that POP1 is only marginally effective in directly suppressing the polymerization of ASC. We also find that both POP2 and POP3 impede the nucleation of the ASC$^{PYD}$ filament.

## Re-examining the Rosetta analyses in light of biochemical experiments

Our biochemical experiments indicated that excess POPs are required to inhibit the polymerization of ASC (*Figure 2E*), which is in agreement with the Rosetta analyses in which no POP•PYD interactions were more favorable than homotypic PYD•PYD interactions (*Figures 1 and 2*). However, although our in silico analyses suggested that POP1 should interact most favorably with ASC$^{PYD}$ (*Figure 1B*), our in vitro and in cellulo experiments consistently showed that POP1 is only marginally inhibitory (*Figure 2*). We thus re-examined our Rosetta results in light of our biochemical experimental results, and noted that the interface energy profiles of POP2•ASC$^{PYD}$ and POP3•ASC$^{PYD}$ are different from that of POP1•ASC$^{PYD}$. For instance, all three POPs contain favorable protein•protein interaction surfaces for ASC$^{PYD}$ (ΔΔG = ΔG$^{PYD•PYD}$- ΔG$^{POP•PYD}$ ≤3.5; arbitrarily determined, marked as blue dots in *Figure 2— figure supplement 3*). However, although both POP2 and POP3 show multiple interfaces with less favorable ΔGs than ASC$^{PYD}$•AS$^{PYD}$ interactions, POP1•ASC$^{PYD}$ interfaces lack such negative interactions (ΔΔG≥10.0; marked as red dots in *Figure 2—figure supplement 3*). These observations in turn raised the hypothesis that a combination of favorable (recognition) and unfavorable interfaces (repulsion) is necessary for POPs to interfere with the assembly of inflammasome PYDs.

## Mechanisms of ALR inhibition by POPs

We then set out to test our amended interpretation of Rosetta analyses on other POP•PYD interactions such as those with AIM2 and IFI16. Here, we noted a mixture of both favorable and unfavorable interactions between all three POPs and both ALRs (*Figure 3—figure supplement 1A–B*), raising the possibility that not only POP3, but the other two POPs might also inhibit the oligomerization of ALRs. To test this, we monitored the filament assembly of AIM2$^{PYD}$-mCherry and IFI16$^{PYD}$-mCherry in HEK293T cells (*Figure 3A–D*). Compared to co-transfecting with eGFP alone, POP1-eGFP reduced the number of AIM2$^{PYD}$ and IFI16$^{PYD}$ filaments, apparently more effective than against ASC$^{PYD}$ (*Figure 2B* vs. *Figure 3B and D*; e.g. at 1200 ng POP1, AIM2$^{PYD}$ and IFI16$^{PYD}$ assemblies were inhibited ~60%, while ASC$^{PYD}$ assembly was suppressed ~20%). On the other hand, co-transfecting POP2 or POP3 essentially obliterated the filament assembly of AIM2$^{PYD}$ and IFI16$^{PYD}$ (*Figure 3A–B and C–D*). In our FRET assays tracking AIM2$^{PYD}$ polymerization, all three POPs decreased the slope of the linear phase in a dose-dependent manner without affecting the initial lag phase (*Figure 3E*; recombinant IFI16$^{PYD}$ does not form filaments in our hands; *Morrone et al., 2014*). Our observations indicate that all three POPs can interfere with the elongation of the AIM2$^{PYD}$ filament, with POP3 being most effective (*Figure 3E*, IC$_{50}$s). Moreover, imaging AIM2$^{PYD}$ filaments using nsEM in the presence of POP1 revealed that the filaments are shorter and fewer, and the presence of POP2/3 abrogated filament formation (*Figure 3F*). As seen from ASC$^{PYD}$, the dearth of filaments in the presence of POP2/3 in our nsEM and in cellulo experiments (*Figure 3B, D and F*) indicated that AIM2$^{PYD}$ oligomers rarely progressed into functional filaments (i.e. rise in FRET signals in *Figure 3E* vs. the lack of filaments in *Figure 3F*).

It is noteworthy that the oligomerization of PYD is important for stable dsDNA binding by ALRs (*Morrone et al., 2015*; *Morrone et al., 2014*). Conversely, although isolated AIM2$^{PYD}$ can form filaments by mass action (i.e. high concentrations; *Morrone et al., 2015*), dsDNA provides a one-dimensional diffusion scaffold to facilitate the assembly of full-length ALRs at significantly lower concentrations (*Morrone et al., 2015*; *Morrone et al., 2014*). AIM2$^{FL}$ forms punctum-like oligomers when transfected in HEK293T cells (*Matyszewski et al., 2021*), and we found that POP1 slightly reduced the number of AIM2$^{FL}$ puncta, while POP2 and POP3 were more effective (*Figure 3—figure*

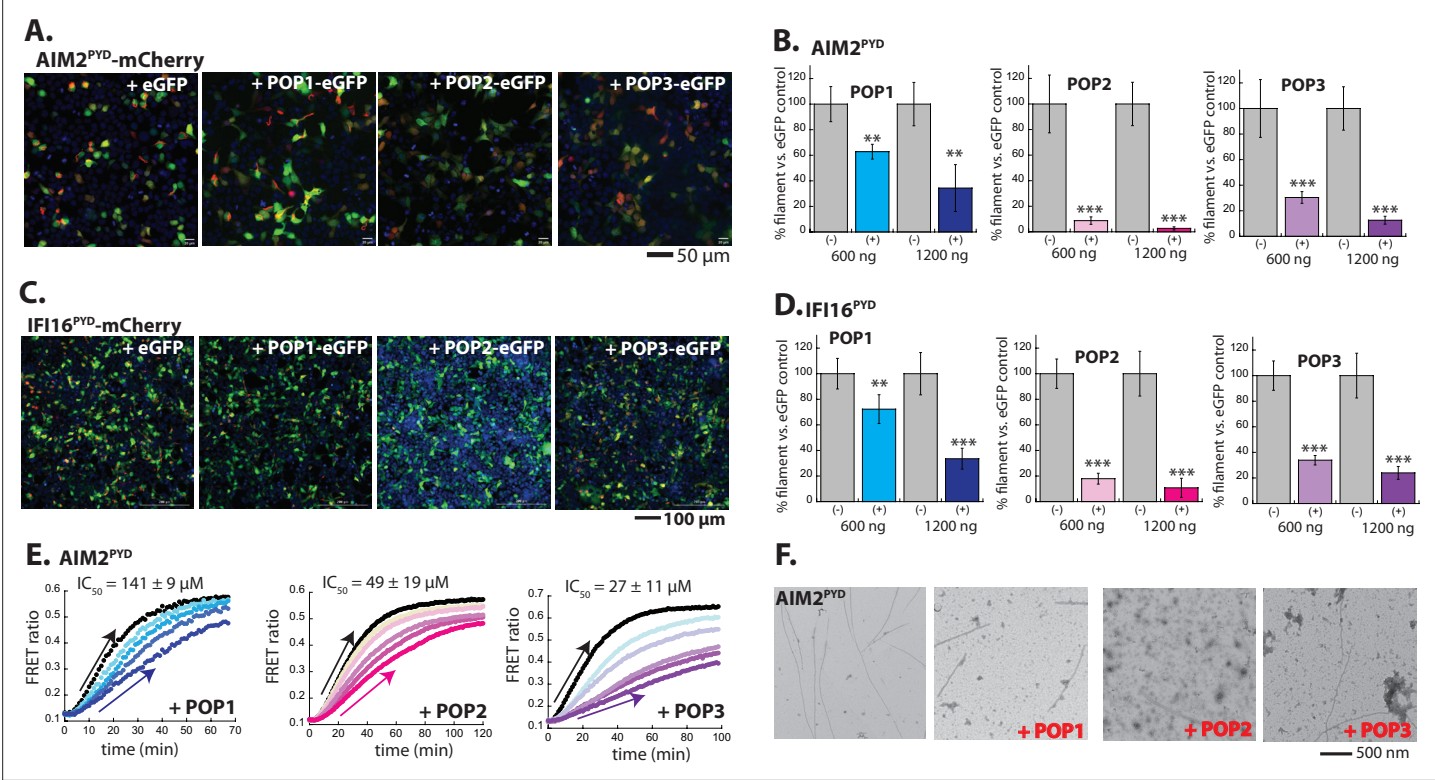

**Figure 3.** Inhibition of AIM2-like receptor (ALR) assembly by pyrin-only-proteins (POPs). (**A**) Sample fluorescent microscope images of HEK293T cells co-transfected with mCherry-tagged AIM2PYD (300 ng; crimson) plus eGFP alone or POP-eGFP (1200 ng; green). Blue: DAPI. (**B**) The relative amounts of AIM2PYD-mCherry filaments (300 ng plasmid) in HEK293T cells when co-transfected with POP-eGFP (+) or eGFP alone (-) (600 and 1200 ng plasmids). N≥4. (**C**) Sample fluorescent microscope images of HEK293T cells co-transfected with mCherry-tagged IFI16PYD (300 ng; crimson) plus eGFP alone or eGFP-tagged POPs (1200 ng; green). Blue: DAPI. (**D**) The relative amount of IFI16PYD-mCherry filaments (300 ng plasmid) in HEK293T cells when co-transfected with POP-eGFP (+) or eGFP alone (-) (600 and 1200 ng plasmids). N≥4. (**E**) Time-dependent increase in Förster resonance energy transfer (FRET) signals of a donor- and acceptor-labeled AIM2PYD (2.5 µM, black) was monitored with increasing concentrations of POP1 (25, 50, 100, and 150 µM), POP2 (12.5, 25, 50, and 75 µM), or POP3 (7.5, 15, 30, 40, and 50 µM); darker shades correspond to increasing POP concentrations. Arrows pointing upper right directions indicate the change (or lack thereof) in the elongation phase in the presence of the highest POP concentrations used. Data shown are representatives of at least three independent measurements (IC$_{50}$s are average values of these experiments. n=3. (**F**) Negative-stain electron microscopy (nsEM) images of AIM2PYD filaments (2.5 µM) in the presence and absence of POP1 (100 µM), POP2 (50 µM), or POP3 (40 µM).

The online version of this article includes the following source data and figure supplement(s) for figure 3:

**Source data 1.** Data values used in plots from *Figure 3* and *Figure 3—figure supplements 1–3*.

**Figure supplement 1.** In silico and in cellulo analyses of POP•ALR interactions.

**Figure supplement 2.** POPs inhibit the dsDNA binding activitiy of ALRs.

**Figure supplement 3.** POPs inhibit ALR oligomerization in cells.

supplement 1C–D). IFI16FL localizes in the nucleus (*Antiochos et al., 2018*; *Kerur et al., 2011*; *Li et al., 2012*), precluding our investigation with cytosolic POPs (*Figure 3—figure supplement 1E*). Consistent with the lack of significant inhibition in cells (*Figure 3—figure supplement 1C–D*), POP1 failed to interfere with dsDNA-binding/oligomerization of recombinant AIM2FL (*Figure 3—figure supplement 2A*). However, POP2 and POP3 still inhibited the dsDNA binding of AIM2, while only POP3 was inhibitory toward the dsDNA binding of recombinant IFI16FL (*Figure 3—figure supplement 2B*). Additionally, imaging experiments with untagged POPs again corroborated our observations using eGFP-tagged proteins (*Figure 3—figure supplement 3*). Overall, our results indicate that POP3 directly inhibits ALR assembly (elongation in particular for AIM2PYD). We also find that POP1 and POP2 can inhibit the assembly of ALR filaments, with POP2 being more effective than the former; the presence of activating ligands can diminish the inhibitory effect of POPs (*Figure 3—figure supplement 2*). Furthermore, these results are consistent with our amended interpretation of Rosetta analyses in

which a combination of favorable and unfavorable interfaces allow POPs to target and inhibit PYD filament assembly.

## POP1 likely targets upstream receptors instead of ASC

Although it has been speculated that POP1 and POP2 would interfere with the recruitment of ASC by NLRPs (*de Almeida et al., 2015*; *Devi et al., 2020*; *Indramohan et al., 2018*; *Periasamy et al., 2017*; *Ratsimandresy et al., 2017*), it remains unknown whether either POP can directly suppress the filament assembly of NLRP[PYD]s. Of note, our investigations here revealed that POP1 is ineffective in inhibiting the oligomerization of ASC (*Figure 2*). Moreover, albeit less inhibitory than POP2 or POP3, POP1 was more effective in suppressing the assembly of AIM2[PYD] and IFI16[PYD] filaments than that of ASC[PYD] (*Figure 3*). These observations strongly suggest that the role of POP1 is to interfere with the assembly of upstream receptors rather than directly inhibiting ASC (i.e. a 'decoy' ASC; targeting ASC or multiple upstream receptors would result in the same phenotype). Indeed, our Rosetta analyses indicated that POP1 can make a combination of favorable and unfavorable interactions with both

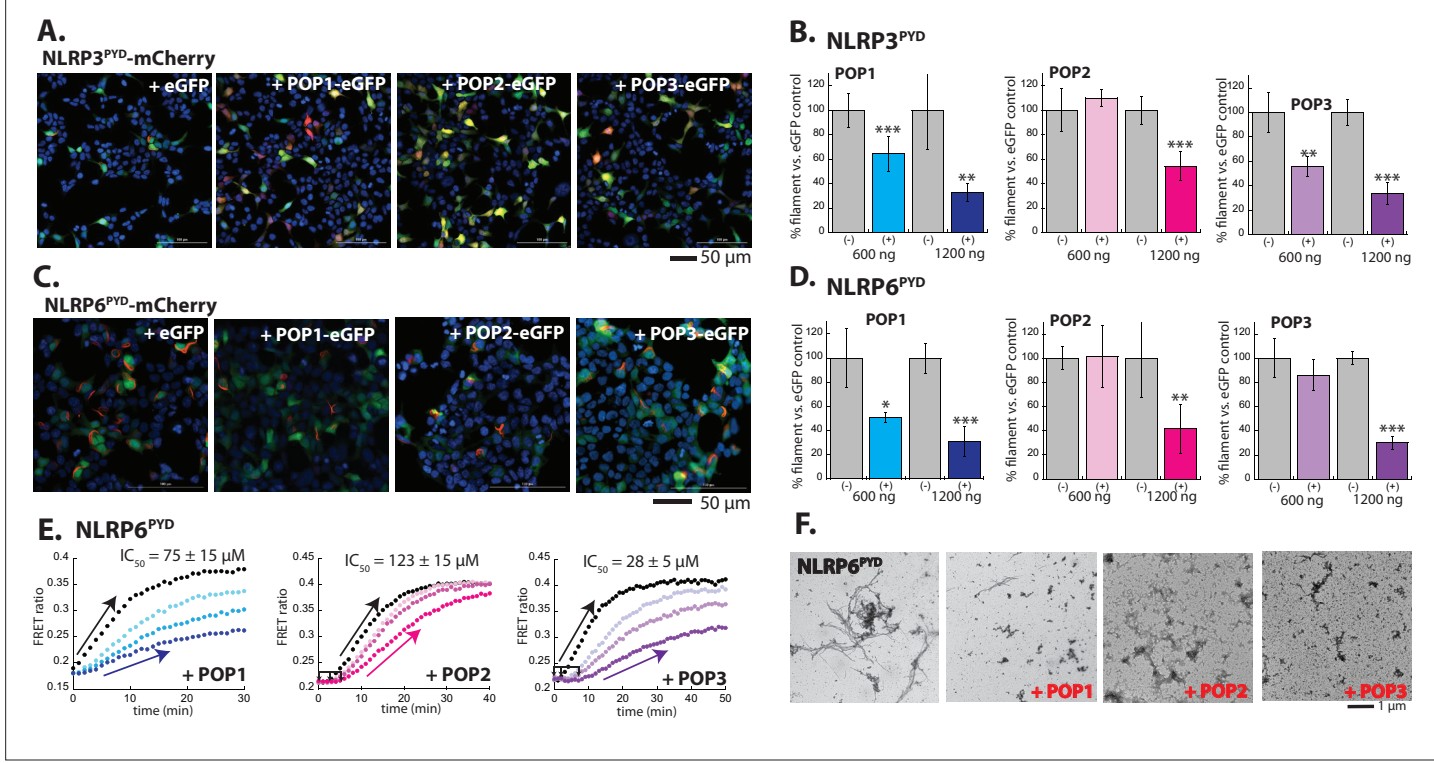

**Figure 4.** Inhibition of NLRP[PYD] assembly by pyrin-only-proteins (POPs). (**A**) Sample fluorescent microscope images of HEK293T cells co-transfected with mCherry-tagged NLRP3[PYD] (600 ng; crimson) plus eGFP alone or eGFP-tagged POPs (1200 ng; green). Blue: DAPI. (**B**) The relative amounts of NLRP3[PYD]-mCherry filaments (600 ng plasmid) in HEK293T cells when co-transfected with POP-eGFP (+) or eGFP alone (-) (600 and 120 ng plasmids). N≥4. (**C**) Sample fluorescent microscope images of HEK293T cells co-transfected with mCherry-tagged NLRP6[PYD] (300 ng; crimson) plus eGFP alone or eGFP-tagged POPs (1200 ng; green). Blue: DAPI. (**D**) The relative amounts of NLRP6[PYD]-mCherry filaments (300 ng plasmid) in HEK293T cells when co-transfected with POP-eGFP (+) or eGFP alone (-) (600 and 120 ng plasmids). N≥4. (**E**) Time-dependent increase in Förster resonance energy transfer (FRET) signals of a donor- and acceptor-labeled NLRP6[PYD] (2.5 µM, black) was monitored with increasing concentrations of POP1 (25, 50, and 100 µM). POP2 (30, 60, and 120 µM). POP3 (15, 30, and 60 µM); darker color shades correspond to increasing POP concentrations. Two- and three-headed arrows indicate the increase in apparent nucleation time. Arrows pointing upper right directions indicate the change (or lack thereof) in the elongation phase in the presence of the highest POP concentrations used. Data shown are representatives of at least three independent measurements (IC$_{50}$s are average values of these experiments. N=3). (**F**) Negative-stain electron microscopy (nsEM) images of NLRP6[PYD] filaments (5 µM) in the presence and absence of POP1 (100 µM), POP2 (30 µM), or POP3 (30 µM).

The online version of this article includes the following source data and figure supplement(s) for figure 4:

**Source data 1.** Data values used in plots from *Figure 4* and *Figure 4—figure supplement 2*.

**Figure supplement 1.** Investigating the interactions between POPs and NLRPs.

**Figure supplement 2.** Experiments for testing the effect of POPs in NLRP oligomerization and signaling.

NLRP3[PYD] and NLRP6[PYD] (*Figure 4—figure supplement 1*). Furthermore, POP2 and POP3 also showed favorable and unfavorable interactions with NLRP3 (*Figure 4—figure supplement 1A*); although the ΔGs between NLRP6 and POP2/3 were largely unfavorable, the Type 3a surface showed an energy score that might allow the two POPs to recognize NLRP6 if present at high enough concentrations (ΔG ~ –9; *Figure 4—figure supplement 1B*, marked as light pink); our reasoning is based on ΔGs ~ –9 seen from native PYD•PYD interactions (e.g. NLRP3[PYD]•NLRP3[PYD] shows ΔGs of ~–8 and –10 on Type 2 and Type 3 interfaces, *Figure 1* and *Figure 4—figure supplement 1A*).

The activation mechanisms of NLRPs are complex and involve different types of active and inactive oligomers (*Andreeva et al., 2021*; *Gong et al., 2021*; *Hochheiser et al., 2022b*; *Lu et al., 2014*; *Ohto et al., 2022*; *Sharif et al., 2019*; *Sharma and de Alba, 2021*; *Shen et al., 2021*; *Shen et al., 2019*); we thus monitored whether POPs impede the filament assembly using the isolated PYDs of NLRP3 and NLRP6, which formed filaments when ectopically expressed in HEK293T cells (*Figure 4A and C*). Of note, NLRP2[PYD]-mCherry did not form filaments when expressed in HEK293T cells (*Figure 4—figure supplement 1C*), precluding further investigations despite its high sequence similarity to POP2 (*Indramohan et al., 2018*; *Periasamy et al., 2017*; *Ratsimandresy et al., 2017*). When co-transfected, POP1 was more effective in suppressing the filament assembly by both NLRP3[PYD] and NLRP6[PYD] than that of ASC[PYD] (*Figure 4A–D*). For example, with 1200 ng POP1, ASC[PYD] assembly was only suppressed by ~20%, but the filament assembly by NLRP3[PYD] and NLRP6[PYD] was suppressed ~60% (*Figure 2B* vs. *Figure 4B and D*). On the other hand, POP2 was less effective in inhibiting the polymerization of NLRP[PYD]s than that of ASC[PYD] (e.g. at 600 ng POP2, ASC[PYD] assembly was abolished, but the assembly of NLRP3[PYD] and NLRP6[PYD] was minimally suppressed; *Figure 2B* vs. *Figure 4B and D*). POP3 was almost equally effective in suppressing the polymerization of NLRP-[PYD]s and ASC[PYD], but not nearly as effective as against ALRs (*Figure 4A–D*). For example, at 600 ng POP3, AIM2[PYD] assembly was suppressed 70%, but those of ASC[PYD] and NLRP[PYD]s were reduced by 40–50%; *Figure 3B* vs. *Figures 2B, 4B and D*. As with other PYDs, cellular imaging experiments using untagged POPs corroborated our observations (*Figure 4—figure supplement 2A–D*).

Next, using FRET donor and acceptor-labeled NLRP6[PYD]s, we then monitored whether POPs suppress the nucleation and/or elongation (recombinant NLRP3[PYD] does not auto-assemble into filaments in our hands [e.g. *Bae and Park, 2011*]). POP1 predominantly inhibited the elongation of NLRP6[PYD] filament, and POP2 appeared to interfere with its nucleation. Although POP3 mostly reduced the elongation kinetics of NLRP6[PYD], it also seemed to interfere with nucleation (*Figure 4E*). Consistent with these observations, the number and length of NLRP6[PYD] filaments were reduced in the presence of POPs (*Figure 4E*). The lack of filaments (*Figure 4F*) despite the increase in FRET signals (*Figure 4E*) again indicates that NLRP6[PYD] oligomers fail to form intact filaments. Finally, we tested the efficacy of POPs in suppressing the downstream signaling activities of inflammasomes by reconstituting NLRP3-dependent IL-18 release in HEK293T cells. Consistent with our imaging and polymerization assays, here, POP2 was most effective in suppressing IL-18 release, POP3 was the second, and POP1 was least effective (*Figure 4—figure supplement 2E*). Our results are consistent with the idea that POP1 acts as a decoy ASC, moderately inhibiting the assembly of upstream PYDs; POP2 is the most potent inhibitor of inflammasomes; POP3 can inhibit inflammasomes beyond AIM2.

## Introducing deleterious mutations for self-assembly can reprogram PYDs into POP-like inhibitors

Next, to further test our working model for understanding the inhibition mechanism of POPs, we generated AIM2[PYD] mutants defective in self-oligomerization and tested whether they can inhibit the assembly of WT (i.e. such mutant proteins contain both favorable and unfavorable interactions for WT). We chose D23K and N73L (*Figure 5A*), because we previously found that not only are these mutations deleterious for self-assembly, but resulting mutant AIM2[FL]•dsDNA oligomers were defective in promoting the polymerization of WT-AIM2[PYD] (*Matyszewski et al., 2021*). Both N73L-AIM2[PYD] and D23K-AIM2[PYD] were impaired in filament assembly (*Figure 5B*, *Figure 5—figure supplement 1A*). Imaging WT-AIM2[PYD] filaments using nsEM in the presence of either mutant showed fewer filaments (N73L) and mesh-like non-filamentous aggregates (D23K; *Figure 5B*). The presence of unlabeled N73L-AIM2[PYD] or D23K-AIM2[PYD] also impeded the polymerization of WT-AIM2[PYD] in our FRET assay, with the former being more effective (*Figure 5C*). Moreover, when co-transfected in HEK293T cells, eGFP-tagged AIM2[PYD] mutants diminished the number of mCherry-tagged WT filaments, again

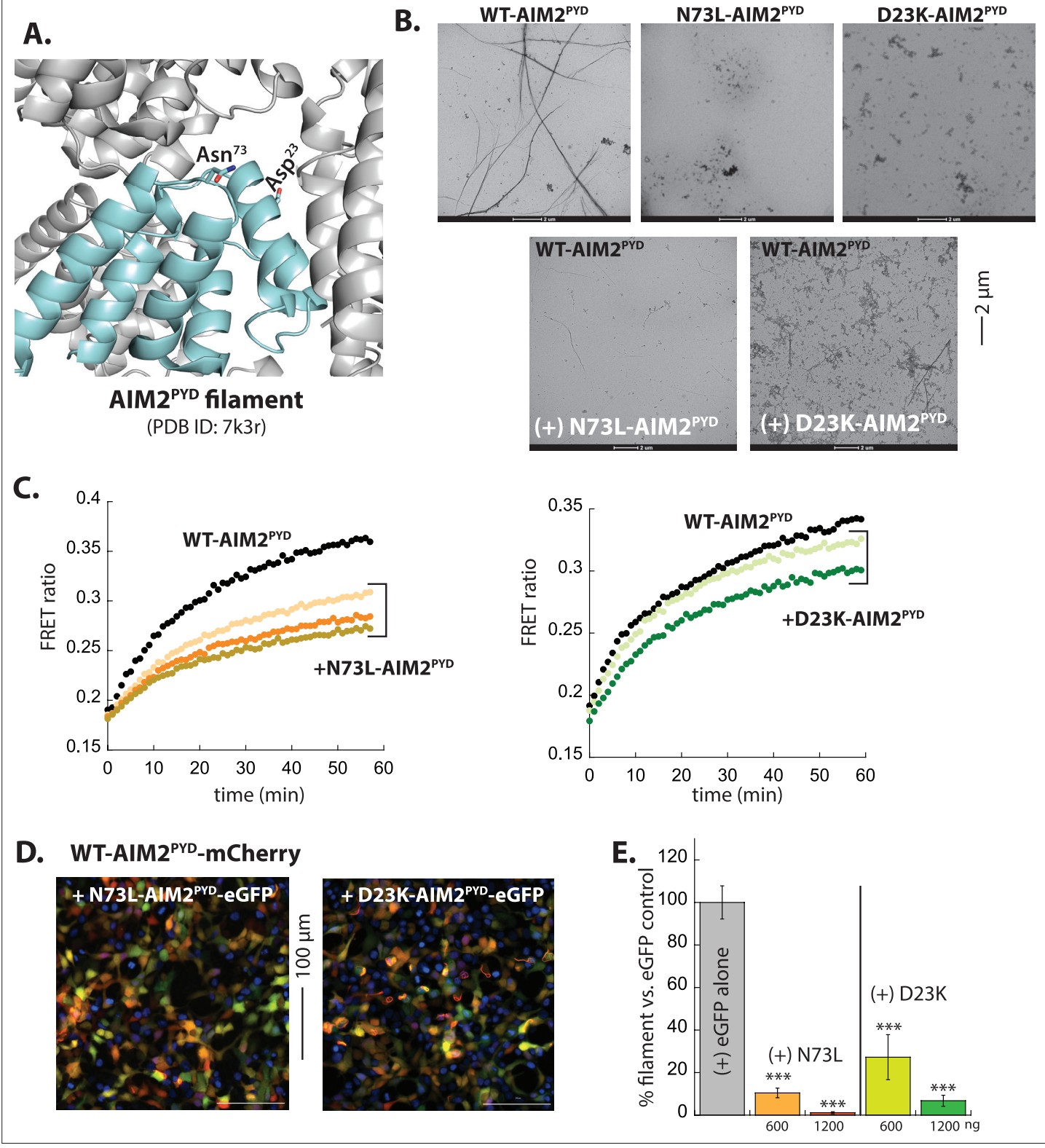

**Figure 5.** Reprogramming AIM2PYD into a pyrin-only-protein (POP)-like inhibitor. (**A**) AIM2PYD filament 'honeycomb' showing the location of mutations. (**B**) Negative-stain electron microscopy (nsEM) images of AIM2PYD variants. WT: 2.7 μM, N73L: 3 μM, D23K: 30 μM. (**C**) Time-dependent increase in Förster resonance energy transfer (FRET) signals of a donor- and acceptor-labeled WT-AIM2PYD (2.5 μM, black) was monitored with increasing concentrations of AIM2PYD mutants (1 (brown), 2 (orange), and 3 (mustard) μM for N73L; 15 (light green) and 30 (dark green) μM for D23K). Data shown are representatives of at least three independent measurements. (**D**) Sample fluorescent microscope images of HEK293T cells co-transfected with

*Figure 5 continued on next page*

*Figure 5 continued*

mCherry-tagged AIM2$^{PYD}$ (300 ng plasmid; crimson) plus eGFP-tagged N73L- or D23K-AIM2$^{PYD}$ (1200 ng plasmids; green). Blue: DAPI. (**E**) The relative amount of AIM2$^{PYD}$-mCherry filaments (300 ng plasmid) in HEK293T cells when co-transfected with N73L or D23K mutants (600 and 1200 ng plasmids). n≥4.

The online version of this article includes the following source data and figure supplement(s) for figure 5:

**Source data 1.** Data values used in plots from *Figure 5*.

**Figure supplement 1.** Reprogramming AIM2PYD as POP-like inhibitors.

N73L being more effective than D23K (*Figure 5D–E*, *Figure 5—figure supplement 1B–C*). Overall, these results consistently support our hypothesis that a combination of favorable and unfavorable interactions underpins the target selection and inhibition by POPs. Our results also provide a proof of principle that PYDs can be reprogrammed into POP-like inhibitors even by a single mutation.

## Discussion
### Redefining the target specificity and inhibitory mechanisms of POPs

A hallmark of inflammasomes is their exceptional stability. For example, ASC promotes 'solidification' of inflammasomes in a prion-like manner, allowing them to perpetuate even after cells undergo pyroptosis (*Cai et al., 2014*; *Franklin et al., 2014*; *Shen et al., 2021*). Moreover, AIM2 and IFI16 filaments also persist and are even stigmatized as autoantigens in debilitating autoimmune disorders such as systemic lupus erythematosus and Sjögren's syndrome (*Antiochos et al., 2018*; *Antiochos et al., 2022*; *Baer et al., 2016*). Indeed, persisting inflammasome oligomers and their aberrant activities are implicated in a wide range of human diseases including COVID-19 (*Karki et al., 2017*; *Tartey and Kanneganti, 2020*; *Vora et al., 2021*). It is thus critical for the host to carefully modulate the assembly of inflammasomes at the onset, as it would be much more difficult to demolish such hyperstable supra-structures. POPs have emerged as major endogenous regulators of inflammasomes by directly interfering with the assembly of PYD filaments, functioning analogous to COPs that target the oligomerization of pro-caspases (*Devi et al., 2020*; *Indramohan et al., 2018*). Although the biological significances of POPs are well established (*de Almeida et al., 2015*; *Devi et al., 2020*; *Indramohan et al., 2018*; *Khare et al., 2014*; *Periasamy et al., 2017*; *Ratsimandresy et al., 2017*), their intrinsic target specificities and inhibition mechanisms have remained speculative.

Our investigations here reveal that POPs interfere with the polymerization (nucleation and/or elongation) of various inflammasome filaments without co-assembling, which is different from COPs that co-assemble into filaments with CARD of caspase-1 (*Lu et al., 2016*). Moreover, also unlike COPs that can inhibit the assembly of pro-caspases at sub-stoichiometric concentrations (*Lu et al., 2016*), excess POPs were necessary to inhibit inflammasome PYDs, especially when an activating ligand was present (dsDNA for AIM2 and IFI16; *Figure 3—figure supplement 2*). Additionally, POP2/3 largely suppressed the nucleation of ASC (e.g. prolonged lags in *Figure 2E*), yet these POPs also interfered with the elongation of upstream receptors (e.g. *Figures 3E and 4E*). Although often considered as a harmful phenomenon, inflammation is integral to host innate defense and survival (*Bennett et al., 2018*; *Meizlish et al., 2021*). We reason that being able to modulate two key assembly steps (nucleation and elongation) while requiring excess POPs is well suited for attenuating inflammasome activities without shutting them down altogether.

Although previous studies showed that POP1 inhibits ASC-dependent inflammasomes (*de Almeida et al., 2022*; *de Almeida et al., 2015*), its target selection and inhibition mechanism remains poorly understood. Here, we provide new insights into the inhibitory mechanism of POP1. That is, POP1 is only marginally effective in inhibiting the polymerization of ASC$^{PYD}$. However, it is likely to modulate inflammasomes largely through interfering with the oligomerization of upstream PYDs, most notably by halting their elongation. It is noteworthy that the expression of POP1 is predominantly induced by IL-1β (*de Almeida et al., 2015*), a major final product of inflammasome cascades (*Broz and Dixit, 2016*; *Zheng et al., 2020*). Thus, it is likely that POP1 is part of a negative feedback loop for attenuating excessive inflammasome activities by preventing perpetuation of upstream filaments (*Figure 6*).

POP2 is thought to function similarly to POP1 by suppressing the activation of ASC (*Periasamy et al., 2017*; *Ratsimandresy et al., 2017*). Interestingly, POP2 is reportedly more effective than POP1

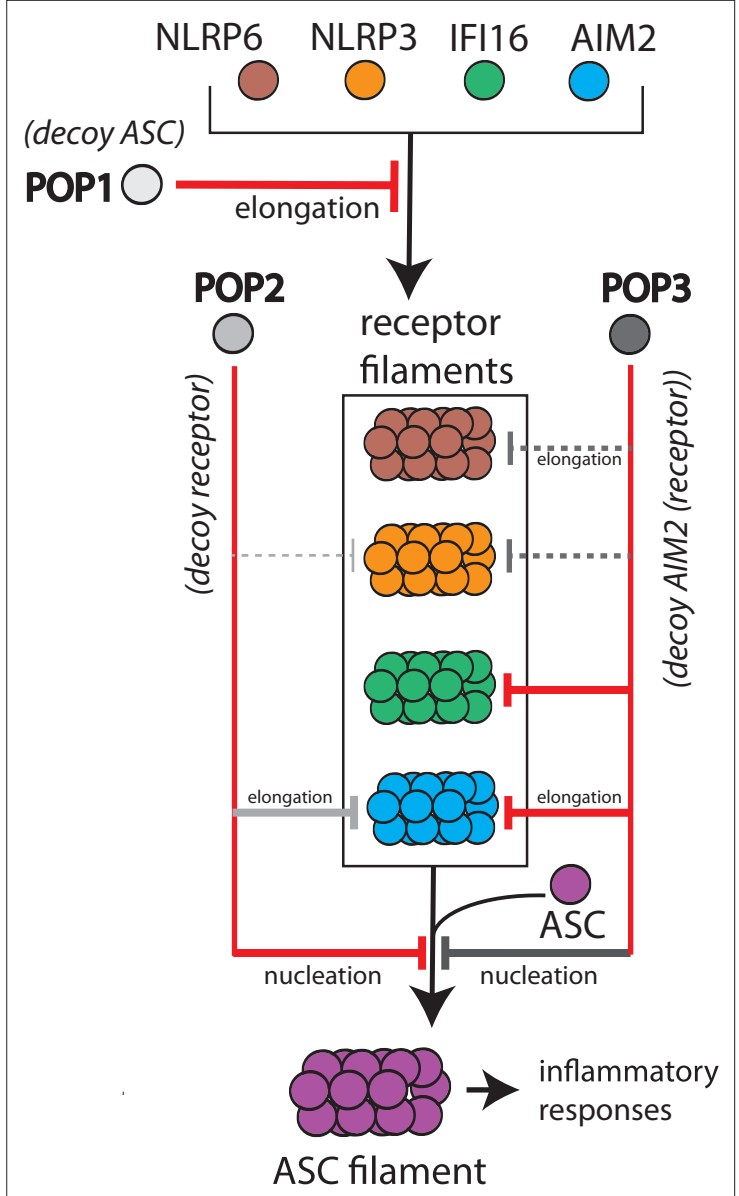

**Figure 6.** Target selection and mode of inhibition of pyrin-only-proteins (POPs). A cartoon summarizing the refined intrinsic target specificity of POPs. Solid red lines indicate the most likely primary inhibitory targets for each POP. Solid gray lines indicate an additional target that each POP could also directly inhibit. Dotted gray lines indicate other possible targets for each POP.

The online version of this article includes the following figure supplement(s) for figure 6:

**Figure supplement 1.** A model for inflammasome assembly by POPs.

in preventing spurious inflammasome activities in vivo (*Periasamy et al., 2017*; *Ratsimandresy et al., 2017*). Nevertheless, it has remained unclear whether it directly suppresses ASC assembly or those of upstream receptors. We find here that POP2 can indeed potently inhibit the nucleation of ASC$^{PYD}$ (*Figure 2*) while moderately suppressing the oligomerization of select upstream receptors such as AIM2. It is noteworthy that the expression of POP2 is readily induced by a wide variety of pro- and anti-inflammatory cytokines (*Ratsimandresy et al., 2017*). Considering that it primarily targets ASC, we postulate that POP2 would act as the pan-inflammasome inhibitor essential for preventing spurious innate immune responses (*Figure 6*; i.e. decoy receptor as it is most similar to upstream NLRP2$^{PYD}$, *Figure 1—figure supplement 1B*).

POP3 was identified as a selective inflammasome inhibitor targeting ALRs (*Khare et al., 2014*). In addition to inhibiting the elongation of the AIM2[PYD] filament, we find here that POP3 can robustly inhibit the nucleation of ASC[PYD] (*Figure 2*), revealing its dual targeting function similar to POP2. We also find that, in principle, POP3 can inhibit the assembly of NLRP[PYD]s (*Figure 4*). Unlike the other two POPs, POP3 is exclusively induced by interferons (IFNs) (*Khare et al., 2014*), a major cytokine family that counteracts IL-1β (*Mayer-Barber and Yan, 2017*). Considering that IFNs also drive the expression of ALRs (*Antiochos et al., 2018*; *Khare et al., 2014*), we postulate that although POP3 is intrinsically capable of inhibiting various inflammasome receptors, the contextual expression would dictate its in vivo targets (*Figure 6*).

## Design principles for inflammasome inhibition by POPs

We envision that the nonequilibrium assembly of inflammasome filaments plays a major role in defining the mechanism of regulation by POPs (i.e. kinetically driven without having conceivable off-rates) (*Cai et al., 2014*; *Matyszewski et al., 2018*). For instance, considering that all POP and PYD protomers share the same overall structure (shape complementarity), we reason that any (semi) favorable protein•protein interaction interfaces would allow POPs and PYDs to associate at least transiently (i.e. classic reversible protein•protein interaction equilibrium). However, the more favorable homotypic PYD•PYD interactions would readily outcompete such meta-stable interactions especially when only basal amounts of POPs are present (*Figure 6—figure supplement 1A*). Moreover, as indicated from the failure to inhibit ASC[PYD] by POP1, if POPs do not contain any strong unfavorable interactions against target PYDs, homotypic PYD•PYD interactions would still outcompete even excess POPs and lock themselves into irreversible filament assembly (*Figure 6—figure supplement 1B*). However, when such meta-stable POP•PYD complexes contain at least one very unfavorable interface, excess POPs would then expose a multitude of adverse protein•protein interaction surfaces that would hamper filament assembly (*Figure 6—figure supplement 1C*). Also of note, given that POP2 and POP3 can form oligomers (*Figure 2—figure supplement 2A–B*), it is highly likely that multimeric POPs are more effective in preventing the association of inflammasome PYDs (*Figure 6—figure supplement 1C*). We found previously that the recognition between AIM2 and ASC occurs when at least one is filamentous (*Matyszewski et al., 2021*). Thus, it is also possible that POPs might preferentially interact with oligomeric PYDs that are not yet fully filamentous (e.g. (pseudo)-nucleation unit), trapping them into nonfunctional states (*Figure 6—figure supplement 1C*).

In closing, a possible caveat of our study is that although our experiments using AIM2[FL], IFI16[FL], and NLRP3[FL] are consistent with those from isolated PYDs (*Figure 3—figure supplements 1C, 2* and *Figure 3—figure supplement 3E*; *Figure 4—figure supplement 2E*), considering that inflammasome receptors contain multiple domains, there could be yet another layer of complexity in POP•inflammasome interactions beyond what we have reported here. It will be also interesting to see to what extent our findings for POP•PYD interactions can be applied to other DD family proteins such as COPs and CARDs. Overall, our multi-disciplinary approach provides an example of how to use in silico predictions judiciously for investigating multipartite protein-protein interactions.

# Materials and methods
## Rosetta simulation

The InterfaceAnalyzer script in Rosetta was used to determine the interaction energy (Rosetta energy units, reu) at individual interfaces of the honeycomb (*Matyszewski et al., 2021*). We used the cryo-EM structures of ASC[PYD] (PDB: 3j63; *Lu et al., 2014*), AIM2[PYD] (PDB: 7k3r; *Matyszewski et al., 2021*), NLRP3[PYD] (PDB: 7pdz; *Hochheiser et al., 2022a*), and NLRP6[PYD] (PDB: 6ncv; *Shen et al., 2019*) filaments to generate corresponding honeycombs. Because the structure of the IFI16[PYD] filament is unknown, we used the eGFP-AIM2[PYD] filament (PDB: 6mb2; *Lu et al., 2015*) that shows a pentameric filament base as a template; using the untagged AIM2[PYD] filament (PDB: 7k3r; *Matyszewski et al., 2021*), which shows a hexameric filament base, as a template resulted in largely unfavorable energy scores (*Figure 1—figure supplement 1C*). For POPs, we used the crystal structure of POP1 (PDB: 4qob), and generated homology models of POP2 and POP3 based on monomeric NLRP3[PYD] (PDB: 7pdz) and AIM2[PYD] (PDB: 7k3r), respectively.

## Cell culture and imaging

Each protein was cloned into pCMV6 vector containing C-terminal mCherry (inflammasome PYDs) or eGFP (POPs). HEK293T cells (ATCC, CRL-11268) were seeded into 12-well plate ($0.1 \times 10^6$ per well) with round cover glass (20 mm). All cells were authenticated via STR profiling and free from mycoplasma. eGFP (or vector) alone or POP-eGFP plasmids (or tagless POPs; 600 and 1200 ng) were co-transfected with inflammasome-mCherry plasmids (300 ng, except NLRP3$^{PYD}$ [600 ng]) at 70% confluence using Lipofectamine 2000 (Invivogen). After 16 hr, cells were washed twice with 1× phosphate-buffered saline, fixed with 4% paraformaldehyde, then mounted on glass slides. Images were taken using the Cytation 5 multi-functional reader equipped with a fluorescent microscope (BioTek) and analyzed via the Gen5 software (BioTek). All paired two-tailed t-tests were performed using Excel (*: $p \le 0.05$, **: $p \le 0.01$; ***: $p \le 0.001$). Source Data are appended for each figure.

## IL-18 release assay

Plasmids encoding full-length NLRP3 (2 μg), ASC (1 ng), pro-caspase-1 (42 ng), and pro-IL-18 (360 ng) were co-transfected into HEK293T cells in the presence and absence of increasing POPs (six-well plate, $0.75 \times 10^6$ per well). After 16 hr, 0.5 $\mu M_f$ nigericin was added to activate NLRP3 and incubated for 4 hr. Supernatant from each well was then collected and the presence of mature IL-18 was monitored by human IL-18 ELISA kit (Invitrogen).

## Recombinant proteins

Inflammasome proteins were generated and labeled with fluorophores when appropriate as previously described (*Matyszewski et al., 2018*; *Matyszewski et al., 2021*; *Morrone et al., 2015*; *Morrone et al., 2014*). POP1 was cloned into the pET21b vector, and POP2 and POP3 were cloned into the pET28b vector containing an N-terminal His$_6$-MBP tag flanked by a cleavage site for TEVp. All recombinant proteins were expressed in *Escherichia coli* BL21 Rosetta2$^{DE3}$ cells and purified using Ni$^{2+}$-NTA followed by size-exclusion chromatography (SEC) (storage buffer: 40 mM HEPES-NaOH at pH 7.4, 400 mM NaCl, 2 mM dithiothreitol, 0.5 mM EDTA, and 10% glycerol). Proteins were then concentrated and stored at –80°C.

## Biochemical assays

FRET and FA-based quantitative assays were conducted as described previously (*Matyszewski et al., 2018*; *Matyszewski et al., 2021*; *Mazanek and Sohn, 2019*; *Morrone et al., 2014*). For example, the polymerization of indicated amounts of FRET-labeled MBP-PYD constructs was triggered by adding TEVp in the presence of increasing amounts of POPs. Half-times for polymerization ($t_{1/2}$s) and the concentration of each POP needed for decreasing the $1/(t_{1/2})$s by 50% ($IC_{50}$) were calculated as described in *Matyszewski et al., 2018*; *Matyszewski et al., 2021*. The $IC_{50}$s for inhibiting the dsDNA-binding activities of ALRs were determined with increasing concentrations of POPs as described previously (*Matyszewski et al., 2018*; *Matyszewski et al., 2021*); the MBP tag was pre-cleaved in these experiments via TEVp for 30 min. Source Data are appended for each figure.

## nsEM

Each PYD was incubated with TEVp for 30 min to remove MBP tag and promote polymerization in the presence or absence of POPs. Samples were then applied to carbon-coated grids and imaged as described previously (*Matyszewski et al., 2018*; *Matyszewski and Sohn, 2019*; *Morrone et al., 2015*).

## Acknowledgements

We thank Dr. Mariusz Matyzewski and Naveen Mohideen for assistance in Rosetta experiments. Fundings from NIH R01GM129342, R35GM145363, NSF MCB1845003 awards to JS NIH K08 AI102696 to MC are acknowledged. Computational Resources were provided by Maryland Advanced Research Computing Center at Johns Hopkins University.

# Additional information

## Competing interests

Zachary Mazanek: Affiliated with Georgiamune. The author has no financial interests to declare. Jungsan Sohn: Reviewing editor, eLife. The other authors declare that no competing interests exist.

## Funding

| Funder | Grant reference number | Author |
| --- | --- | --- |
| National Institute of General Medical Sciences | R01GM129342 | Shuai Wu |
| National Institute of General Medical Sciences | R35GM145363 | Shuai Wu |
| National Science Foundation | MCB1845003 | Archit Garg |
| National Institute of Allergy and Infectious Diseases | K08AI102696 | Aubrey Roland |

The funders had no role in study design, data collection and interpretation, or the decision to submit the work for publication.

## Author contributions

Shuai Wu, Data curation, Formal analysis, Validation, Investigation, Methodology, Writing – review and editing; Archit Garg, Christina M Stallings, Jacob Lueck, Data curation, Investigation, Writing – review and editing; Zachary Mazanek, Data curation, Validation, Methodology; Gretchen Belotte, Jeffery J Zhou, Data curation, Formal analysis, Investigation; Aubrey Roland, Investigation; Michael A Chattergoon, Formal analysis, Project administration; Jungsan Sohn, Conceptualization, Supervision, Funding acquisition, Investigation, Writing – original draft, Writing – review and editing

## Author ORCIDs

Shuai Wu ![ORCID] http://orcid.org/0000-0003-1677-0221
Archit Garg ![ORCID] http://orcid.org/0000-0002-5931-2522
Michael A Chattergoon ![ORCID] https://orcid.org/0000-0002-2220-1116
Jungsan Sohn ![ORCID] https://orcid.org/0000-0002-9570-2544

## Decision letter and Author response

Decision letter https://doi.org/10.7554/eLife.81918.sa1
Author response https://doi.org/10.7554/eLife.81918.sa2

# Additional files

## Supplementary files

• MDAR checklist

## Data availability

Source data filles for Figures 2, 3 and 4 contain the numerical data used to generate figures.

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
