## [Editor Report]

This important work proposes general principles by which a central class of inflammatory mediators called inflammasomes are regulated by "pyrin only" proteins. The authors employ a convincing combination of in silico and biochemical measurements in support of their model. The work is of broad interest to those interested in the biochemical regulation of inflammatory signaling pathways.

---

## [Decision Letter]

**Decision letter after peer review:**

Thank you for submitting your article "Design Principles for Inflammasome Inhibition by Pyrin-Only-Proteins" for consideration by *eLife*. Your article has been reviewed by 3 peer reviewers, and the evaluation has been overseen by a Reviewing Editor and Carla Rothlin as the Senior Editor. The reviewers have opted to remain anonymous.

The reviewers have discussed their reviews with one another, and the Reviewing Editor has drafted a consensus set of essential revisions. Overall our judgement is that the paper is of potential enough importance to be considered for *eLife*, but significant revisions including new experiments are required.

In this manuscript, Mazanek and colleagues combine computational analysis and in vitro experiments to develop a comprehensive analysis of the ability of pyrin-only proteins (POPs) to inhibit inflammasome assembly. The pyrin domain (PYD) of each POP has a total of six interaction surfaces that can bind to the PYD of different inflammasome components. The results lead the authors to propose that a mixture of favorable and unfavorable interaction surfaces is required for a POP to inhibit a given inflammasome component. The results are potentially important as they may provide a generalizable mechanism by which POPs inhibit inflammasome activation. However, additional experimentation is required to fully justify the authors' model. Indeed, their 'mixed interface' model is not directly tested and the possibility that POPs could inhibit inflammasome signaling by being incorporated into filaments is not ruled out. A limitation of the study is that the authors only measure PYD filament formation and do not actually measure downstream inflammasome activation.

Essential revisions:

1. The authors show that the MBP tag affects the oligomerization of POPs. The POPs used in Figures 2A, 3A, and 4A contain a GFP tag which may change the inhibitory effect of POPs on ASC filament formation. Experiments with untagged POPs are therefore required to validate the results.

2. The authors take the reduction of PYD filamentation as an indication of inhibition, but it is not clear how they ruled out the possibility that POP1 co-assembles into the ASCPYD filaments and inhibits inflammasome formation by repressing the recruitment of Caspase-1 (as POP1 lacks the CARD the effector domain). Thus, some functional assays measuring downstream Caspase-1 activation are required. In addition, the possibility that POP1 and ASC co-assemble could be tested directly with FRET experiments in which one protein is the donor and the other is the acceptor. Without such experiments, the statement in the Discussion "Our investigations here reveal that POPs interfere with the polymerization (nucleation and/or elongation) of various inflammasome filaments without co-assembling, … " does not appear to be justified.

3. Further computational analysis should be performed to determine if the theory that a combination of favorable and unfavorable interactions is generally applicable. Does this theory account for other PYD/PYD interactions and CARD/CARD interactions? For example, for the AIM2PYD/ASCPYD interface, do they see only a favorable interface or a mixture? How about two unrelated PYDs, such as between AIM2PYD and NLRP3PYD? How about for COPs? How about the homotypic interfaces between the POPs themselves? In addition, as raised by reviewer 3, the authors do not consider at all in this manuscript that there is a seventh interface described for PYDs besides the hexagonal assembly in the filament. This is the homodimer interface seen, e.g., for NLRP3, in the crystal structure of the PYD and in size exclusion chromatography (Bae and Park, 2011; 3QF2). For completeness, the energy scores should be also calculated for this interface. It might well be that POPs associate in this binding mode as heterodimeric assemblies to monomeric PYDs of NLRs or ALRs to regulate their activities. This would be somehow reminiscent of profilin binding to actin, regulating the pool of free G-actin for filament assembly.

4. To more directly test the mixed-interaction model, the authors should use their Rosetta structural predictions as a guide to introduce mutations into the various POP1/ASC-PYD interfaces to reduce the binding energies of those specific interactions and then test whether the introduction of a single or multiple, weak interactions then allows POP1 to restrict ASC-PYD oligomerization. To further elucidate their mixed interface model, the authors should also address whether the weak interactions need to be on the same 'half' of the interface, e.g., does weakening the 1b and 2b interfaces lead to better disruption of PYD filamentation than a 1b/1a combination mutant?

*Reviewer #1 (Recommendations for the authors):*

1. Figures 2A, 3A, and 4A could be complemented with untagged POPs, which may limit the observation of POP expression, but could still tell how they affect ASC filamentation in their physiological form by observing the mCherry signal.

2. Figure 2-4, panel Es and panel Fs: It should be described in the figure legend that the MBP-tagged ASC protein is used to measure the FRET signal upon TEV cleavage. It should also be noted whether the MBP-tagged POPs are used in these assays, or the proteins w/o tag.

3. To test the co-assembly model, the FRET assay should be performed with either POP1 or PYD as the donor, and the other protein as the receptor.

4. An inflammasome functional assay should be performed to test if POP1 inhibits inflammasome activation w/o disrupting ASC filament formation.

5. If the authors can calculate the AIM2PYD /ASCPYD interface, do they see only a favorable interface or both? How about two unrelated PYDs, such as between AIM2PYD and NLRP3PYD? How about COPs?

*Reviewer #2 (Recommendations for the authors):*

I believe that this is a strong manuscript with a careful and detailed analysis of the ability of the POPs to disrupt PYD filament formation. However, I believe that there are two critical points that the authors should address.

First, the authors propose that the mix of strong and weak interactions between the POPs and the various PYDs regulate the ability of the POPs to block PYD filamentation. For example, the authors propose that the reason POP1 fails to restrict ASC-PYD filamentation is due to a lack of weak interactions, whereas both POP2 and POP3 are able to interfere with ASC-PYD oligomerization through a mix of strong and weak interactions. This is an interesting hypothesis, however I believe that it needs to be tested directly.

In particular, I would like to see the authors use their Rosetta structural predictions as a guide to introduce mutations into the various POP1/ASC-PYD interfaces that reduce the binding energies of those specific interactions and then test whether the introduction of a single, or multiple, weak interactions then allows POP1 to restrict ASC-PYD oligomerization. Additionally, I believe that it may be prudent to generate combinations of mutants along the different interfaces to uncover whether the weak interfaces are required to be on the same facing to disrupt PYD oligomerization – for example, would weakening the 1b and 2b interfaces lead to better disruption of PYD filamentation than a 1b/1a combination mutant? I believe these experiments are necessary for the authors to demonstrate that a combination of strong and weak interfaces is necessary to disrupt PYD filamentation as they propose. Additional mutagenesis of the other POP/PYD interactions may be an appropriate alternative to directly test the proposed mechanism.

A potential area for clarification on this point is the mechanism through which POP1 fails to regulate ASC PYD oligomerization. Looking at the del(G) calculated for POP1/ASC-PYD interaction the del-del(G) compared with ASC-PYD/ASC-PYD is only ~1. Compared with the POP2/NLRP3-PYD interaction in which POP2 fails to prevent NLRP3-PYD oligomerization and where the del-del(G) is nearly 30, this seems unexpected. Is POP1 becoming incorporated within the ASC-PYD filament? If the authors mix individually labeled POP1 and ASC-PYD protein in their FRET assay do they observe co-oligomerization and or co-localization in their spec assays?

The second issue I believe that the authors need to address is whether the POPs truly disrupt inflammasome signaling through the mechanisms they propose. While I appreciate the authors' microscopy demonstrating a lack of PYD spec formation in their reconstituted system I believe that they need to demonstrate that the POPs are able to restrict inflammasome activation in a more natural setting. For example, this could be accomplished by transducing the POPs (and/or mutants that are able or unable to restrict PYD oligomerization) into THP1s, prime, and activating the NLRP3 inflammasome and showing a loss of cell death and/or cytokine processing. Of course, it would be outside the scope of this manuscript to expect this for all the inflammasomes and POPs in combination, however, I believe that showing POP-mediated inhibition for at least one inflammasome/POP pair, along with POP mutants to demonstrate the loss of specific interface interactions proposed by the Rosetta calculations are required for the POPs to block inflammasome activity.

*Reviewer #3 (Recommendations for the authors):*

It seems to this reviewer that the authors have written the manuscript in chronological order as they have performed the experiments, which might, however, not be the best way to present their data. I propose to rewrite and reorganize the manuscript, to better make the points. The Discussion is a bit repetitive and seems overwritten. Some shortening, despite adding new ideas and considerations (see below), might be reasonable.

This reviewer does not understand the scientific rationale regarding the PYD sequence identity and similarity of POPS to either NLRs or ALRs as a measure of their potential regulatory/inhibitory function in the respective inflammasome formation. Think of a protein with a single-point mutation that would abrogate filament formation. It would have the exact same deltaGs in five interfaces but only one is different. If this protein would be added, e.g., in a 1:1000 ratio to wt protein that has been pre-incubated to form filaments, it might stop elongation. This is, for example, how some capping proteins work in the actin cytoskeleton. Thus, the immanent interpretation of the Rosetta calculations of the ASC-POP1 interfaces in Figure 1 showing that POP1 is so similar to ASC-ASC interfaces would be that POP1 could be easily integrated into the ASC filament as a surrogate. I, therefore, disagree with the authors about the statement of the "counterintuitive"-ness: "Moreover, although somewhat counterintuitive at first, further examining Rosetta in silico results in light of our initial biochemical experiments suggested that a combination of favorable and unfavorable interactions underpin the target specificity of POPs." [Discussion section, page 22 middle, and similar in the Results]. A more balanced consideration here would be favorable.

In addition to this comment, from the sequence similarity of the PYDs of NLRP3, AIM2, or NLRP6 to ASC-PYD we learn that the degree of sequence homology does not determine or correlate to the ability for homotypic transition and filament elongation.

Figure 1 is not a good start, particularly, as it is repeated and reversed later in Figure 2 Supplement 2, Figure 3 Supplement 1, and Figure 4 Supplement 1. I suggest putting the entire present Figure 1 into the supplement and rather start with something similar to Figure 2, Supplements 2, and so on, which is more informative. I also wonder if it wouldn't be easier to show the subtractions of the deltaGs (so the δ(deltaGs)) to better visualize the differences in the Rosetta calculations. The idea with the blue and red dots is good, the boundaries (<3.5 dG and >10 dG) seem reasonable, so I would include them from the beginning. Thinking of it, an ideal inhibitor might have 5 blue dots and one red. This would be preferentially in interface I (the largest) on the b-side (the growing direction) and it would ensure assembly but prevent elongation. I suggest to revise the text in light of this consideration.

The abundance of proteins is a major determinant in the law of mass action, besides the thermodynamic energies of the direct contacts made. The transcriptional control and the protein levels of the POPs might be therefore an important determinant similar to the yields of NLRs and ALRs. The authors reflect on this in the Discussion ("…, excess POPs were necessary to inhibit inflammasome PYDs, especially when an activating ligand was present, …) but the more than 1 stoichiometry required is puzzling and rather points towards a "chaperone-type" function for POPs.

The authors do not consider at all in this manuscript, that there is a seventh interface described for PYDs besides the hexagonal assembly in the filament. This is the homodimer interface seen, e.g., for NLRP3, in the crystal structure of the PYD and in size exclusion chromatography (Bae and Park, 2011; 3QF2). For completeness, the energy scores should be also calculated for this interface. It might well be that POPs associate in this binding mode as heterodimeric assemblies to monomeric PYDs of NLRs or ALRs to regulate their activities. This would be somehow reminiscent of profilin binding to actin, regulating the pool of free G-actin for filament assembly. This might be indeed an effective way to regulate effector signaling.

Discussion: "Our investigations here reveal that POPs interfere with the polymerization (nucleation and/or elongation) of various inflammasome filaments without co-assembling, … " Where is this shown, is it from the "%filament" in Figures 2B, D or the level of saturation in Figure 2E? Further: "…, excess POPs were necessary to inhibit inflammasome PYDs, especially when an activating ligand was present …". Isn't this a strong argument to consider also other interfaces/binding modes than the filament assembly. See the previous comment on profilin-G-actin binding.

[Editors’ note: further revisions were suggested prior to acceptance, as described below.]

Thank you for resubmitting your work entitled "Design Principles for Inflammasome Inhibition by Pyrin-Only-Proteins" for further consideration by *eLife*. Your revised article has been evaluated by Carla Rothlin (Senior Editor) and a Reviewing Editor and all three of the original reviewers.

The reviewers remain interested in the work and consider that it is potentially appropriate for *eLife*. The reviewers are satisfied with many of the revisions and in particular agree that you should not be made to extend your observations to CARD-CARD interfaces. However, because you failed to address all the "required" revisions, significant issues remain.

In particular, after substantive discussion among the reviewers, there remains considerable skepticism about the fundamental model that is being proposed. This skepticism is summarized by one reviewer as follows: "The authors propose that a mixture of favorable and unfavorable interactions is critical in mediating inhibition of filament formation, however, it is unclear if this prediction holds true. For example, looking at Figure 4 and the associated supplemental material the authors find that POP1, POP2, and POP3 can inhibit NLRP6(PYD) filamentation. However, the IC50 varies by nearly 6-fold and more confusingly looking at the energy calculations in Figure S4.1 there does not seem to be a significant difference to explain why this would be. Additionally, if I compare the δ-deltaGs in the NLRP6 calculation with those from AIM2 calculation I cannot see any clear reason why POP3 is the strongest inhibitor of NLRP6 when every interaction is energetically unfavorable compared with the other POP/NLRP6 or NLRP6 homo-interactions. Yet, looking at the inhibition of AIM2, POP3 has the fewest energetically unfavorable interactions but is still the strongest inhibitor. The calculations for POP homotypic interactions supplied in the reviewer response further confound this issue – POP1 shows favorable interactions to form filaments similar to the free energies calculated for ASC and NLRP3, whereas POP2 does not – and yet as the authors acknowledge, POP2, but not POP1 forms filaments. In short, I am not convinced that there is sufficient evidence that the calculated energies are predictive of the behavior of the different POPs… without any experiments that directly address whether the energy calculations are predictive of POP-mediated inhibition, any connection between the calculated Rosetta energies and inhibition are correlative." The other reviewers tend to concur with this assessment. In order for your paper to be considered for publication we require the following:

1. You need to explicitly recognize/discuss/acknowledge that NLRs and ALRs contain additional domains beyond the PYD that can modify the energy landscape to create oligomers, and it is vital to consider this aspect as a caveat to your study which focuses on dissociated PYD interactions.

2. The consensus of the reviewers remains that you need to test your model by employing site-directed mutagenesis. This was a clear requirement in the previous decision letter (point 4), and the reviewers were not convinced by your argument that this is not necessary. One reviewer suggests: "One could easily imagine replacing a favorable interaction with alanine to gently weaken the interaction or with glutamate or other charged residue to break the interface. By taking POP1 and introducing mutations to disrupt one or multiple surfaces the authors could show that these mutants do or do not inhibit filament formation ASC. Alternatively, the authors could introduce mutations into the ASC PYD and determine whether the mutant protein could now act like a POP and block WT ASC filament formation. While I appreciate that doing such a comparison for every POP and PYD in the paper would be excessive, I strongly believe that the authors need to experimentally test their model with at least some site-directed mutants".

---

## [Author Response]

Essential revisions:1. The authors show that the MBP tag affects the oligomerization of POPs. The POPs used in Figures 2A, 3A, and 4A contain a GFP tag which may change the inhibitory effect of POPs on ASC filament formation. Experiments with untagged POPs are therefore required to validate the results.

We added new results monitoring the oligomerization of mCherry-tagged PYDs, ASCFL, and AIM2FL in the presence of untagged POPs, which remain consistent with our existing results (Figure supplements following Figures 2-4).

2. The authors take the reduction of PYD filamentation as an indication of inhibition, but it is not clear how they ruled out the possibility that POP1 co-assembles into the ASCPYD filaments and inhibits inflammasome formation by repressing the recruitment of Caspase-1 (as POP1 lacks the CARD the effector domain). Thus, some functional assays measuring downstream Caspase-1 activation are required. In addition, the possibility that POP1 and ASC co-assemble could be tested directly with FRET experiments in which one protein is the donor and the other is the acceptor. Without such experiments, the statement in the Discussion "Our investigations here reveal that POPs interfere with the polymerization (nucleation and/or elongation) of various inflammasome filaments without co-assembling, … " does not appear to be justified.

We included data showing that POP1 is least effective in inhibiting the NLRP3/nigericin-mediated release of mature interleukin-18 (Figure 4—figure supplement 2E; i.e., a functional assay requested by the reviewer). Here, we also find that POP2 is most effective, likely by directly suppressing ASC polymerization. Our observation here is also consistent with the report by Ratsimandresy et al. reporting that POP2 is more effective in suppressing ASC/inflammasomes than POP1.

We also generated fluor-labeled recombinant POP1 and confirmed that there are no discernable FRET signals between donor-labeled POP1 and acceptor-labeled ASCPYD, again supporting that POP1 and ASCPYD do not intermix (Figure 2, Figure Supplement 2C).

3. Further computational analysis should be performed to determine if the theory that a combination of favorable and unfavorable interactions is generally applicable. Does this theory account for other PYD/PYD interactions and CARD/CARD interactions? For example, for the AIM2PYD/ASCPYD interface, do they see only a favorable interface or a mixture? How about two unrelated PYDs, such as between AIM2PYD and NLRP3PYD? How about for COPs? How about the homotypic interfaces between the POPs themselves?

We are afraid that the reviewers are asking for future orthogonal studies significantly beyond the scope of the current work. This present manuscript focuses on PYD•POP interactions by studying more than a dozen pairwise interactions using three different methods (*in silico*, recombinant proteins, in cell imaging). We strongly believe that any additional *in silico* predictions need to be validated and (re)interpreted in light of biochemical experiments. Thus, investigating different systems such as CARD•CARD and NLRP•ASC interactions with our rigorous approach would require years’ worth of additional work, especially if they have unique patterns (rules) for themselves. Also of note, there are reports on the co-activation of different inflammasome receptors (e.g., Han et al., Sci. Immun., 2021). We do not yet know whether they directly interact with/regulate one another via PYDs or only communicate with ASC. We refrain from suggesting any interactions or lack thereof without biological/biochemical studies (i.e., beyond the scope of the present manuscript).

In our recent work (Matyzewski et al., Nat Com, 2021), we did not see any strong unfavorable energy scores between AIM2 and ASC, and our *in silico* approaches helped in identifying the directional interaction between AIM2PYD and ASCPYD filaments. Of note, we do not treat Rosetta energy scores as absolute free energy terms (as they are not), we use them as relative-yet-quantitative measures to guide our investigations. Importantly, our conclusion then was strictly based on our subjects at hand, as is here (any potential differences found in investigating NLRP•ASC interactions would not validate or invalidate our working model for AIM2•ASC).

We ran Rosetta interface energy analyses on hypothetical homotypic interactions for each POP. Although we refrain from making any claims without conducting extensive biochemical studies, it appears that POP1 lacks the symmetric landscape for “top” and “bottom” halves seen from PYD filaments, which like reflect the lack of filament formation.

On the other hand, POP2 and POP3 show significantly unfavorable energy scores vs. PYD filaments such as AIM2PYD and ASCPYD. Of note, we do not yet understand how POP2 and POP3 form oligomers. For instance, we do not know whether they oligomerize via different interfaces than those mediate filament assembly in PYDs (Type 1-3). We believe that delineating how POP2 and POP3 form oligomers is beyond the scope of our current manuscript and requires extensive combinations of *in silico* and biochemical experiments.

**Author response image 1. sa2fig1:** 

We believe our work here opens a door for testing to what extent our findings reported here are applicable to other death-domain (DD) proteins (or other filamentous assemblies).We look forward to such future studies to compare and contrast, improve, and even modify our understanding of how DD proteins interact with one another. To further clarify our stance, we added the following sentence at the end of the Discussion:

“Future investigations using molecular dynamics simulations and extensive mutagenesis will further delineate the complexity of oligomerization mechanisms and target specificities of POPs in more detail. It will be also interesting to see to what extent our findings for POP•PYD interactions can be applied to other DD family proteins such as COPs and CARDs. Overall, our multi-disciplinary approach provides an example of how to use in silico predictions judiciously for investigating multipartite protein-protein interactions.”

In addition, as raised by reviewer 3, the authors do not consider at all in this manuscript that there is a seventh interface described for PYDs besides the hexagonal assembly in the filament. This is the homodimer interface seen, e.g., for NLRP3, in the crystal structure of the PYD and in size exclusion chromatography (Bae and Park, 2011; 3QF2). For completeness, the energy scores should be also calculated for this interface. It might well be that POPs associate in this binding mode as heterodimeric assemblies to monomeric PYDs of NLRs or ALRs to regulate their activities. This would be somehow reminiscent of profilin binding to actin, regulating the pool of free G-actin for filament assembly.

In the report by Bae and Park, a minor population of the NLRP3PYD dimer was observed in the SEC/MALS analysis using very high protein concentrations under an acidic buffer system without any reducing agents (~600-800 µM protein at pH 5.0; Bae and Park, JBC, 2011). Considering the conditions, we are afraid that such dimer formation is unlikely physiologically important. Indeed, we regret that we do not find any biological relevance for NLRP3 dimers in the literature (full-length or PYD), and even Bae and Park did not report any functional relevance of this dimer in their paper. Although this 2011 study revealed the structure of NLRP3PYD monomer, it predates the groundbreaking discovery by Hao Wu, Ed Egelman, and colleagues showing that inflammasomes form filaments (Lu et al., Cell, 2014).

Nevertheless, we generated dimer models of PYDs and POP•PYDs based on the NLRP3PYD dimer crystal structure and conducted Rosetta interface analyses. We regret that we do not see any compelling signs indicating that such hypothetical interactions would play a major role in the target specificity of POPs:

**Author response table 1. sa2table1:** Rosetta interface energy scores between PYD•PYD and PYD•POP interaction on the putative dimer seen from the crystal structure of NLRP3^PYD^.

	Self	POP1	POP2	POP3
ASC^PYD^	-6.6	-5.6	-5.2	-4.4
AIM2^PYD^	-4.0	2.1	1.3	0.8
IFI16^PYD^	46.6	-5.6	2.6	10.8
NLRP3^PYD^	-17.2	-8.6	-3.7	-15.1
NLRP6^PYD^	-6.6	0.1	-5.6	-3.3

Also importantly, the putative dimer interface is part of the filament interface we have already included in our analyses (Author response image 2; type 3a/1b). We thus respectfully disagree that this is a uniquely important seventh interface.

**Author response image 2. sa2fig2:** The NLRP3PYD filament. The colored region is thought to mediate dimerization.

4. To more directly test the mixed-interaction model, the authors should use their Rosetta structural predictions as a guide to introduce mutations into the various POP1/ASC-PYD interfaces to reduce the binding energies of those specific interactions and then test whether the introduction of a single or multiple, weak interactions then allows POP1 to restrict ASC-PYD oligomerization. To further elucidate their mixed interface model, the authors should also address whether the weak interactions need to be on the same 'half' of the interface, e.g., does weakening the 1b and 2b interfaces lead to better disruption of PYD filamentation than a 1b/1a combination mutant?

We considered such mutagenesis approaches early on but decided against them. For example, one can transplant POP2 residues on POP1, and *vice versa*. However, this would simply make POP1 more like POP2 and *vice versa*, which, in our view, do not provide significant new insights.

Instead, we are developing a new approach of combining the *Monte Carlo* simulation with Rosetta to further delineate how POPs and PYDs interact (this project has been inspired by our prior work reported in Matyzewski et al., PNAS, 2018). Briefly, our approach will test the probability of assembling the filament base (and inhibition of) by various PYD-POP pairs with different number of favorable and unfavorable interfaces (we will also give different weights to each interface and introduce *in silico* mutations). As with the current manuscript and our prior studies, any *in silico* predictions will be tested using biochemical methods employing recombinant proteins and cellular assays, which will take more than a year to complete. This follow-up study will help in further improving our understanding (model) of how POPs regulate PYDs. Moreover, it could also allow us to generate “designer” POPs that can target a wide variety of PYDs. We look forward to reporting our findings in the future.

We would also like to stress that our manuscript presents a comprehensive approach for investigating the interaction between POPs and PYDs. Moreover, the present manuscript marks only the beginning of our long-term goal of elucidating the “interaction codes” that underpin the specificity and interaction mechanisms of DD proteins.

Reviewer #3 (Recommendations for the authors):It seems to this reviewer that the authors have written the manuscript in chronological order as they have performed the experiments, which might, however, not be the best way to present their data. I propose to rewrite and reorganize the manuscript, to better make the points. The Discussion is a bit repetitive and seems overwritten. Some shortening, despite adding new ideas and considerations (see below), might be reasonable.

Thanks for the suggestion. We shortened the Discussion.

Additionally, our apologies for the oversight, we improved the resolution of Figure 1—figure supplement 1.

This reviewer does not understand the scientific rationale regarding the PYD sequence identity and similarity of POPS to either NLRs or ALRs as a measure of their potential regulatory/inhibitory function in the respective inflammasome formation.

We do not intend to support or refute the rationale behind this previously proposed notion by Devi et al., Indramohan et al., de Almeida et al., and others (cited throughout the manuscript); however, our observations here indicate it’s indeed more complex.

[Editors’ note: what follows is the authors’ response to the second round of review.]

1. You need to explicitly recognize/discuss/acknowledge that NLRs and ALRs contain additional domains beyond the PYD that can modify the energy landscape to create oligomers, and it is vital to consider this aspect as a caveat to your study which focuses on dissociated PYD interactions.

We added such a disclaimer at the end of the paper.

2. The consensus of the reviewers remains that you need to test your model by employing site-directed mutagenesis. This was a clear requirement in the previous decision letter (point 4), and the reviewers were not convinced by your argument that this is not necessary. One reviewer suggests: "One could easily imagine replacing a favorable interaction with alanine to gently weaken the interaction or with glutamate or other charged residue to break the interface. By taking POP1 and introducing mutations to disrupt one or multiple surfaces the authors could show that these mutants do or do not inhibit filament formation ASC. Alternatively, the authors could introduce mutations into the ASC PYD and determine whether the mutant protein could now act like a POP and block WT ASC filament formation. While I appreciate that doing such a comparison for every POP and PYD in the paper would be excessive, I strongly believe that the authors need to experimentally test their model with at least some site-directed mutants".

We added new mutagenesis data at the end of the *Results section*. Briefly, we introduced mutations that hamper the self-assembly of AIM2^PYD^ (i.e., unfavorable) and found that resulting mutant proteins can inhibit the polymerization of WT-AIM2^PYD^.